# Tuning without Peeking: Provable Privacy and Generalization Bounds for LLM Post-Training

## Abstract

Gradient-based optimization is the workhorse of deep learning, offering efficient
and scalable training via backpropagation. However, exposing gradients during
training can leak sensitive information about the underlying data, raising privacy
and security concerns such as susceptibility to data poisoning attacks. In contrast,
black box optimization methods, which treat the model as an opaque function,
relying solely on function evaluations to guide optimization, offer a promising
alternative in scenarios where data access is restricted, adversarial risks are high,
or overfitting is a concern. This paper introduces *BBoxER*, an evolutionary black-
box method for LLM *post-training* that induces an information bottleneck via
implicit compression of the training data. Leveraging the tractability of informa-
tion flow, we provide non-vacuous generalization bounds and strong theoretical
guarantees for privacy, robustness to data poisoning attacks, and extraction attacks.
In experiments with LLMs, we demonstrate empirically that black-box optimiza-
tion methods—despite the scalability and computational challenges inherent to
black-box approaches—are able to learn, showing how a few iterations of *BBoxER*
improve performance, generalize well on a benchmark of reasoning datasets, and
are robust to membership inference attacks. This positions *BBoxER* as an attractive
add-on on top of gradient-based optimization, offering suitability for deployment
in restricted or privacy-sensitive environments while also providing non-vacuous
generalization guarantees.

## 1 Introduction

Large Language Models (LLMs) have revolutionized natural language processing, achieving strong
performance across diverse tasks through training on massive datasets – often hundreds of billions of
tokens – and large-scale architectures (Kaplan et al., 2020; Wei et al., 2022a). Yet LLMs suffer from
persistent weaknesses, foremost among them vulnerability to extraction attacks (Haim et al., 2022;
Carlini et al., 2021; 2024) where training data can be leaked (Zhu et al., 2019; Hitaj et al., 2017),
raising potential privacy concerns. They are also vulnerable to data poisoning (Wan et al., 2023;
Miranda et al., 2025), and lack meaningful generalization guarantees at scale. These challenges are
amplified in settings where data is scarce, sensitive, or adversarially curated, underscoring the need
for more robust and privacy-aware optimization methods.

Black-Box Optimization (BBO) covers methods suited to settings where gradient information is
unreliable or unavailable (e.g., non-differentiable or noisy objectives). These algorithms are nat-
urally robust to local minima and non-convexity. Although around for decades, BBO has been
recognized only recently in machine learning, becoming a core tool in AutoML (Hutter et al., 2019),
for hyperparameter tuning, algorithm selection (Feurer & Hutter, 2019), and neural architecture
search (NAS) (White et al., 2023). Beyond random and grid search, techniques like Evolutionary
Algorithms (e.g., CMA-ES (Hansen & Ostermeier, 1996; 2003; Hansen, 2023), Differential Evolu-
tion (Storn & Price, 1997), Particle Swarm Optimization (Kennedy & Eberhart, 1995)) and Bayesian
Optimization (Močkus, 1975; Mockus, 1981; Garnett, 2023) offer diverse trade-offs in scalability,
sample efficiency, and exploration-exploitation dynamics. BBO methods still face limitations: they
scale poorly to high-dimensional parameter spaces and require many function evaluations, making
them unsuitable for full model training (except for Deep Reinforcement Learning, due to its high
parallelism (Salimans et al., 2017)). To reconcile the strengths and weaknesses of both paradigms,

we adopt a *hybrid strategy*: using gradient-based pretraining for scale, followed by BBO on a small, targeted subset of the model to tackle privacy, poisoning, and overfitting.

Post-training introduces its own risks: instruction tuning and generation can lead to exploitative behavior (Shu et al., 2023; Huang et al., 2024), overfitting may harm robustness and generalization (Yang et al., 2024), and data poisoning can occur during reinforcement learning from human feedback (RLHF) (Wang et al., 2024; Baumgärtner et al., 2024) or direct preference optimization (DPO) (Rando & Tramèr, 2024; Pathmanathan et al., 2025). This setting is well-suited for retrofitting – a concept initially developed to refine word vectors post-hoc using semantic knowledge (Faruqui et al., 2015), which we generalize here to LLMs for arbitrary downstream tasks.

We introduce *BBoxER* (Black-Box Evolutionary Retrofitting), a comparison-based black-box retrofitting method (Videau et al., 2024c) applicable after pretraining, fine-tuning, or reinforcement learning loops such as GRPO (Shao et al., 2024). Positioned in the outermost layer of the pipeline, *BBoxER* requires no gradient access and integrates seamlessly with existing black-box libraries and algorithms (Rapin & Teytaud, 2018). The resulting sequence of queries and updates forms an implicit compression trace, enabling the use of compression-based generalization theory (Campi & Garatti, 2023; Jiang et al., 2022), currently the only route to non-vacuous generalization bounds for LLMs (Lotfi et al., 2024). It allows us to derive strong generalization and privacy guarantees that - unlike VC dimension (Vapnik & Chervonenkis, 1971) or Rademacher complexity (Bartlett & Mendelson, 2002) - do not depend on the number of modified parameters but on the complexity of the optimization trajectory. As a result, *BBoxER* ensures *privacy by design*, robustness to data poisoning, robustness to extraction attacks, and provable, *non-vacuous* generalization bounds to control overfitting. It is uniquely positioned to provide improvements from aggregate feedback of billions of users anonymously, without accessing individual data.

While our approach establishes strong theoretical guarantees, guarantees alone would be meaningless if they came at the cost of utility—after all, doing nothing also ensures perfect privacy. What is striking with *BBoxER* is that, even under tight query budgets of only a few hundred model evaluations, it consistently improves performance and generalization on billion-scale LLMs across reasoning benchmarks. In other words, the framework delivers *provable generalization and safety without sacrificing usefulness*, turning theory into a practical tool for post-hoc adaptation of LLMs, especially in privacy-sensitive settings.

**Our contributions.** *BBoxER* introduces a new perspective on LLM post-training: instead of fine-tuning weights or relying on reinforcement learning, we show that simple black-box retrofitting can achieve safe and modular adaptation. Building on this idea, we make three main contributions:

  • **A general-purpose retrofitting framework.** We formalize *BBoxER* (Alg. 1), a comparison-based black-box optimization scheme that compresses optimization traces, enabling safe and modular adaptation of pretrained and post-trained LLMs (Sec. 3).

  • **Strong theoretical guarantees.** We derive non-vacuous generalization bounds (Cor. 1) that scale linearly with dataset size, and establish *privacy by design* through provable robustness to poisoning (Equation (7)) and resistance to extraction attacks (Cor. 2) (Secs. 4, 4.2 and 4.3).

  • **Empirical validation on billion-scale LLMs.** Despite tight query budgets, we show consistent gains on GSM8K and related math benchmarks with Llama3.1-8B and Qwen-2.5-3B, and provide empirical evidence supporting our theoretical claims: retrofitted models with *BBoxER* resist Membership Inference Attacks, unlike fine-tuned counterparts at equal utility (Sec. 5).

## 2  BACKGROUND AND RELATED WORK

An extensive exposition of related work is provided in App. A.

**Generalization Bounds for LLMs.** Classical generalization theory based on uniform convergence and VC dimension fails for large models like LLMs. Modern approaches either apply PAC-Bayes bounds using compression techniques (Zhou et al., 2018; Lotfi et al., 2022; 2024) or adopt algorithmic stability (Bousquet & Elisseeff, 2000), which does not rely on hypothesis class complexity. Our approach aligns with the latter by leveraging comparison-based optimization and using union-bound-like arguments (e.g., Bonferroni correction) to derive non-trivial generalization bounds.

**Data Poisoning.** Recent poisoning attacks show that corrupting as little as 1–5% of fine-tuning or preference data can induce persistent, targeted behaviors in LLMs (Baumgärtner et al., 2024; Wang et al., 2024), even in quantized variants (Egashira et al., 2024). These attacks evade standard detection and persist during inference. Moreover, the common strategy of training on public data then fine-tuning on private data is also vulnerable to backdoor injection (Feng & Tramèr, 2024). Specifically, Feng & Tramèr (2024) showed that allowing neurons to retain a gradient from a single input and later "deactivate" to avoid overwriting makes models susceptible to gradient inversion. Our black-box optimization approach is, by design, secure against such attacks. While theoretical bounds exist for simpler models (Steinhardt et al., 2017), robust guarantees for LLMs remain elusive.

**Privacy and Robustness.** LLMs risk leaking private or confidential content (Carlini et al., 2021). Differential privacy (DP) techniques (Dwork et al., 2006) like DP-SGD (Abadi et al., 2016; Flemings et al., 2024) and PATE (Papernot et al., 2016) are the standard paradigm for privacy preserving learning, offering formal guarantees but suffering from utility and scalability challenges, due to the necessary per-sample gradient clipping and adding noise that is inversely proportional to $\varepsilon$. Our method avoids the need for noisy gradient updates by learning directly from user preference signal, in the specific case of A/B testing, ensuring perfect privacy of users input prompts and outputs. Unlike SFT or RLHF, BBoxER ensures that the model never accesses any raw example, only aggregated preference signals. This aligns with emerging concerns in LLM safety (Hayes et al., 2024; Barbero et al., 2025) about leakage during alignment, rather than database style privacy.

## 3 METHODS: RETROFITTING WITH BBO

Retrofitting (Douglas, 2006; Dawson, 2007; Dixon & Eames, 2013) is routinely deployed in industry, in order to adapt old devices to new contexts or new data. In machine learning, the term has mainly been used in natural language processing since Faruqui et al. (2015), who adapted the representation of the word vector to take semantic knowledge into account. It is performed after classical training, i.e., without retraining the entire network. Videau et al. (2024c) adapted retrofitting to the black-box optimization context. In the present paper, we adapt retrofitting to LLM post-training and instantiate it as *BBoxER* and consider comparison-based black-box optimization algorithms to obtain the explicit compression properties that lend themselves to the strong bounds proved in Sec. 4.

A key motivation for using BBO is that it allows post-training with non-differentiable loss functions such as accuracy or signals from A/B testing. And since it only requires model inference, it significantly reduces memory usage. More importantly, the main motivation of this work is to leverage the compression achieved by reducing the entire dataset's forward pass to just a single—or a few—comparison results, yielding a finite branching factor. This constrains our deployment of BBO to comparison-based algorithms, which include Evolutionary Algorithms (EA) and Strategies (ES) like Population-Based Evolution Strategies (e.g., CMA, DiagonalCMA), Differential Evolution, and Particle Swarm Optimization - all described in App. D.2. While a vast class of BBO algorithms is not *comparison based* (see App. D.2 for an anthology of BBO) this nonetheless gives us a substantial variety of BBO algorithms to deploy. For example, the (1+1)-OneFifth strategy samples new points from a Gaussian centered on the current best solution and adjusts the mutation step size based on the acceptance rate. Similarly, CMA-ES updates both the mean and covariance of its sampling distribution based on comparisons within the current population. More algorithms are described in App. D. Our *BBoxER* framework is modular and algorithm agnostic, and we instantiate it with methods from the Nevergrad suite (Rapin & Teytaud, 2018).

### 3.1 THE RETROFITTING FRAMEWORK: *BBoxER*

**Notation.** Let $\mathcal{D}$ be the class of datasets of size $s$ over a domain $\mathcal{X} \times \mathcal{Y}$. Models map $\mathcal{X}$ to $\mathcal{Y}$. Let $\omega$ be the random seed corresponding to the possibly randomized optimization algorithm $a$ and $\omega_D$ the random seed associated with the draw of $D \in \mathcal{D}$.

**Initial model and modifiers.** We denote $m_0$ the initial model, irrespective of its creation. $modified(m_0, x)$ refers to a modification of $m_0$ parametrized by some $x$. We prove all results for an arbitrary $modified$ function. For example, $modified$ can refer to a multiplicative low-rank update of a particular attention layer of the LLM, $x$ then contains the two low-rank matrices that parametrize the update (see App. B).

Alg. 1 presents our abstract retrofitting framework *BBoxER*.

---

**Algorithm 1** *BBoxER*$(m_0, \omega, D, a, b)$: Returns a modification of initial model $m_0$ by BBO algorithm $a$ on dataset $D$ with budget $b$ and random seed $\omega$.

---

1:  $I_0 = a.initialize(\omega)$
2:  **for** $i \in \{1, \ldots, b\}$ **do**
3:      $m_i = modified(m_0, x_i := a.ask())$
4:      $k_i := a.numCases()$
5:      $choice_i = a.compare((m_1, \ldots, m_i), D) \in \{1, \ldots, k_i\}$
6:      $a.tell(choice_i)$
7:      $I_i = (I_{i-1}, choice_i)$
8:  **end for**
9:  $FinalModel = modified(m_0, \widehat{x} = a.recommend())$

---

First, Algorithm $a$ is initialized and returns its initial internal state $I_0$ in Line 1. Then for $b$ iterations, Algorithm $a$, via $a.ask$ at Line 3, proposes a new sample $x_i$, which used to obtain a new model $m_i$ through $modified$, which can be any function that modifies the model weights using the sampled noise by Algorithm $a$. Examples of such $modified$ function are provided in App. B. Line 4 introduces the branching factor $k_i$, representing the finite number of possible outputs of $a.compare$, which we assume is upper bounded. At Line 5, $a.compare$ returns $choice_i \in \{1, \ldots, k_i\}$, the outcome of comparing various previously proposed models $m_1, \ldots, m_i$ on the data $D$. For instance, the comparison may involve the current model and the best model obtained so far, which corresponds to $k_i = 2$. The outcome of this comparison can be based on user-provided preference, such as in A/B testing, or on performance evaluations over $D$, where we compute a metric (e.g., accuracy) but only retain the binary comparison result (e.g., 0 or 1). In Line 7, internal state of $a$ is updated to account for the comparison result. Finally $a.recommend$ at Line 9 returns the modification $\widehat{x}$ used to obtain the final model. A step by step execution example of *BBoxER* can be found in App. K.

It is important to note that the internal state $I_b$ of algorithm $a$ is entirely determined by $\omega, a, b, choice_1, \ldots, choice_b$ and contains all the information needed to obtain $\widehat{x}$, and thus the final modified model. Moreover, the number of distinct possible outputs of *BBoxER* is upper bounded by the number of distinct possible values of $I_b$. We assume that the randomness of algorithm $a$ is fixed given the seed $\omega$; stochastic algorithms can be obtained by randomizing the seed $\omega$ in $a$.

### 3.2   *BBoxER*: Link with compression

In Alg. 1, $FinalModel = modified(m_0, \hat{x})$ is a deterministic function of $m_0, \omega, a, b, D$. And since $\widehat{x}$ only depends on $\omega, a, b, choice_1, \ldots, choice_b$, we can construct $\widehat{x}$ without $D$. Hence, there exists a mapping X

s.t.
$$\widehat{x} = X(\omega, a, b, choice_1, \ldots, choice_b) \tag{1}$$

But $(choice_1, \ldots, choice_b)$ is a deterministic function of $m_0, \omega, a, b, D$. Hence, there exists a deterministic $compression$ function $c$ defined by

$$c(m_0, \omega, a, b, D) := (choice_1, \ldots, choice_b) \tag{2}$$

Here, $X$ mimics the *BBoxER* Alg. 1 but returns $\widehat{x}$ instead of $modified(m_0, \widehat{x})$, and uses $c$ instead of $D$. The dataset, replaced by $c = (choice_1, \ldots, choice_b)$, is entirely removed from Equation (1). To flesh out *BBoxER* as compression, we can merge Equation (1) and Equation (2) into:

$$FinalModel = modified(m_0, X(\omega, a, b, c := compression(m_0, \omega, a, b, D))) \tag{3}$$

$FinalModel$ only depends on the data through the compression bottleneck $c$ of bounded size (see Figure 2, Appendix, for illustration). The impact of this compression is formalized in Sec. 4, and used to prove guarantees in terms of overfitting, privacy robustness w.r.t poisoning and extraction attacks.

## 4   Theoretical analysis

Analyzing *BBoxER* within a unified framework through its information bottleneck that compresses the dataset into a sequence of comparisons, allows us to state and prove strong and precise bounds on its ability to preserve generalization, its robustness to poisoning, its privacy guarantee, and its robustness against extraction attacks. For mathematical background see App. C, additional results in App. E, and proofs in App. G.

## 4.1 Overfitting bounds

In this section, we adress generalization by bounding the gap between empirical and true loss in terms of the variability of the algorithm's internal state. Notably, these bounds are governed by the optimization path, with no dependence on the number of model parameters optimized during the retrofitting process.

Let $L(x) = L(modified(m_0, x))$ denote the true loss over dataset distribution $\mathcal{D}$, and $\widehat{L}(x) = \widehat{L}(modified(m_0, x))$ the empirical loss on sampled dataset $D$. Let $N(\omega, a, b)$ denote the number of possible internal states over all possible datasets in $\mathcal{D}$, then from Line 5 in Alg. 1 we have:

$$N(\omega, a, b) \leq \sup_{D \in \mathcal{D}} \prod_{i=1}^{b} k_i(\omega, D, a, b). \tag{4}$$

We fix the algorithm $a$ seed $\omega$ (the stochastic case is discussed in App. E.2) and study how the number of distinct possible internal states affects generalization. We defer the general case Thm 1 to App. E, showing that the generalization gap is controlled by the number of reachable configurations in Equation (4). It also provides a uniform bound over all intermediate models $(m_1, \ldots, m_b)$, ensuring that empirical losses remain close to the true risk throughout optimization. The bound is obtained using mild assumptions, only requiring that the that empirical loss is a good proxy for the true loss, a standard concentration-bound hypothesis in classical learning theory (van der Vaart & Wellner, 1996; Vapnik, 1995) (see App. C.1).

We present below the practical case where the individual losses are bounded in [0,1], as with accuracy or user preference signals $\in \{0, 1\}$, and where data points are sampled independently from each other.

**Corollary 1.** *(informal) Under standard concentration assumptions (Equation (A.1)), if individual losses are in $[0, 1]$ and Hoeffding's inequality applies (Eq. A.2) with dataset size s, then:*

$$P_{\omega_D}\left(|\widehat{L}(\widehat{x}) - L(\widehat{x})| \geq \epsilon\right) \leq 2 \cdot \left(\prod_{i=1}^{b} k_i\right) \cdot \exp(-2s\epsilon^2) = 2 \cdot \exp\left(\ln 2 \cdot \sum_{i=1}^{b} \log_2(k_i) - 2s\epsilon^2\right) \tag{5}$$

Most BBO algorithms used within *BBoxER* bound the average $k_i$ by 2 or 3. Interpreting $\log_2(k_i)$ as a budget in bits means that to avoid overfitting we can afford a number of iterations $b$ proportional to $s$ divided by the bits used per step. Precisely, if we assume $\forall i, k_i = 2$, then for an error gap $\epsilon$ and confidence $\delta$, we have from Cor. 1 the following formula (numerical values can be found in App. E.7):

$$P_{\omega_D}\left(|\widehat{L}(\widehat{x}) - L(\widehat{x})| \geq \epsilon\right) \leq \delta \Leftrightarrow b \leq \frac{\ln(\delta) + 2s\epsilon^2}{\ln(2)} - 1 \tag{6}$$

**Bounds on additional BBO algorithms.** Our analysis extends to a wide range of evolutionary BBO methods. In particular, App. E.4 shows that so called *Bet-And-Run* variants - running $N$ parallel runs each with a budget of $b/N$ - can reduce overfitting and scale the total budget while preserving generalization. App. E.5 and App. E.6 provide analogous bounds for Population-Based Evolution Strategies (e.g., CMA, DiagonalCMA), Differential Evolution, and Particle Swarm Optimization.

## 4.2 Privacy and poisoning

We now consider the two key risks associated with model post-training: (a) *poisoning*, where a small number of corrupted inputs can steer the outcome, and (b) *privacy leakage*, where individual training samples or - in the case of A/B testing - individual user preferences - might be inferred from the final model. Standard fine-tuning methods, including privacy-aware approaches such as differentially private stochastic gradient descent (DP-SGD), rely on batch-level gradient signals that inherently encode information on the underlying data. *BBoxER* on the other hand, relies only on aggregate values over the entire dataset, namely on the outcomes of model comparisons $(choice_1, \ldots, choice_b)$. Each $choice_i$ represents a compressed signal derived from the full dataset $D$. For a training dataset $D$, in this context we define *preference signals* to be the sample-level decisions that lead to aggregate decision $choice_i$ over $D$. In the case of A/B testing on prompt/output pairs, these would be the binary user preferences that aggregate to a model preference. In the case of more

general metrics, the preference signal for a sample is defined as the result of models comparison over the subdataset {sample}.

To analyze *BBoxER*, we consider a simplified setting, where $\forall i, k_i = 2$, and in which two models $\{0, 1\}$ are compared by majority vote over $s$ independent preference signals $r_i \in \{0, 1\}$. The observed frequency is $f = \frac{1}{s} \sum_{i=1}^{s} r_i$. An adversary may corrupt up to $m$ values among the $r_i$, yielding a perturbed frequency $f'$ such that $|f' - f| \leq m/s$. This follows the standard poisoning model (Kearns & Li, 1993), where the adversary is computationally unbounded and fully informed, but constrained in the number of values it can alter. Lem. 1 in App. G bounds the probability that such perturbations alter the comparison outcome.

Using a union bound argument, we extend this analysis to *BBoxER* (Alg. 1), which performs $b$ consecutive comparisons. In iteration $j \in \{1, \ldots, b\}$, the algorithm collects binary preferences $r_{j,1}, \ldots, r_{j,s}$, aggregates them into $f_j = \frac{1}{s} \sum_{i=1}^{s} r_{j,i}$, and selects a model based on whether $f_j > 1/2$. Under poisoning, the adversary may shift $f_j$ to $f'_j$ with $|f'_j - f_j| \leq m/s$, independently at each round. The following theorem bounds the impact of these perturbations on the final output.

**Theorem 2** (Robustness of *BBoxER* under poisoning). *Let* `FinalModel = modified`$(m_0, \hat{x})$ *denote the selected model using decisions* $choice_j = \text{sign}(f_j - \frac{1}{2})$. *Let* $output_1$ *denote the model chosen under unperturbed frequencies* $f_j$, *and* $output_2$ *under perturbed frequencies* $f'_j$ *satisfying* $|f'_j - f_j| \leq m/s$. *Then:*

$$P(output_1 \neq output_2) \ \leq \ b\frac{2m + 1}{\sqrt{2s\pi}}. \tag{7}$$

**Privacy guarantee** Let $D_1$ and $D_2$ be two datasets that differ by a single sample. We examine the effect of this difference on the FinalModel given by *BBoxER*, and we distinguish two cases:

• **privacy of users prompts and outputs** (like in A/B testing) We assume that the preference signals are provided directly by users, and that the datasets $D_1$ and $D_2$ can differ on the prompts or outputs. Since *BBoxER* relies solely on users preference signals to perform model comparisons, and does not access prompts or outputs, privacy is ensured by design and we have $(\varepsilon, \delta) = (0, 0)$. In this scenario, the user preference signal isn't considered private, but the input prompts and outputs are private and they can be completely different without impacting the output of BBoxER.

• **algorithm stability** We assume that the $D_1$ and $D_2$ can differ on an entire sample. This is the case when the preference signal needs to be *computed* from the data before aggregating and passing it to the algorithm or when we consider the user's preference signal to be a private information in A/B testing. In the following we bound the influence of one sample. From Equation (7) using $m = 1$ (only 1 sample difference) we get that : $P(output_1 \neq output_2) \leq 3b/\sqrt{2\pi s}$. Which translate to $(\varepsilon, \delta) = (0, \frac{3b}{\sqrt{2s\pi}})$ for differential privacy (DP), but remains an insufficient bound given that $\delta$ needs to be polynomial in inverse dataset size $1/s$ for DP(Vadhan, 2017).

Theorem 2 yields two immediate consequences: First, unless $m = \Omega(\sqrt{s})$, the retrofitting framework is resilient to poisoning attacks, and second in A/B testing case, privacy is guaranteed *by design*, covering prompts and outputs without additional mechanisms. These points are further discussed in App. F. It is important to note that, while other privacy preserving methods such as DP-SGD (Flemings et al., 2024) enforce privacy by perturbing gradients and consequently suffer from utility degradation as $\varepsilon$ becomes small, *BBoxER*'s compression-based guarantees remain robust. These structural differences highlight a fundamental advantage of our approach.

### 4.3 ROBUSTNESS TO EXTRACTION ATTACKS

Prior work shows that training data can sometimes be extracted from models (Carlini et al., 2021; Henderson et al., 2023), even in privacy-preserving settings like federated learning. We formalize vulnerability to data extraction through identifiability: if one can reliably infer which dataset was used during fine-tuning, the process may be considered vulnerable.

Formally, let $m_0$ be fine-tuned on unrelated datasets $D_1, \ldots, D_n$. For text, "unrelated" can mean differing by one word out of five while remaining coherent. A dataset $D$ with $B$ bits may yield $n \approx 2^{B/k}$ such variants by altering one bit per $k$. We define vulnerability to data extraction as follows:

**Definition 1** (simple formalization of vulnerability to data extraction). *An algorithm $A$ producing $A(m_0, D)$ from model $m_0$ and dataset $D$ is vulnerable to data extraction for $\mathcal{D} = (D_1, \ldots, D_n)$ (all distinct) if there exists a mapping $E$ such that*

$$\forall i \in \{1, \ldots, n\}, \ E(A(m_0, D_i)) = i. \tag{8}$$

This criterion can be tested practically: generate $n$ unrelated variants $D_1, \ldots, D_n$, fine-tune models on each $D_i$, and check whether each model assigns the highest likelihood to its own training set. If so, it indicates vulnerability to data extraction. We argue this is unlikely for retrofitting, due to limited capacity to encode training data. Thm 5 in App. G.2 formalizes this, leading to the corollary below:

**Corollary 2.** *For given $\omega, a, b$ s.t. $k_i \leq 2$ for all $i$,* BBoxER *is not vulnerable to extraction for $\mathcal{D} = (D_1, \ldots, D_n)$ if $n > 2^b$. More generally,* BBoxER *is not vulnerable for $\mathcal{D}$ if $n > \prod_i k_i$.*

Thus, if the number of alternative datasets exceeds that of distinguishable outputs, no extractor can identify the training data. Since $2^b$ grows only exponentially with budget $b$, while the number of unrelated textual variants grows exponentially with dataset size (e.g., $> 400\text{B}$ bits for Wikipedia articles, $> 20\text{M}$ for *War and Peace*), extraction is very unlikely in practice.

## 5 EXPERIMENTAL RESULTS: LEARNING WITH PRIVACY AND ROBUSTNESS

The theoretical bounds in Sec. 4 establish that comparison based BBOs in the framework of *BBoxER* offer strong generalization guarantees and inherent privacy. However, theoretical guarantees alone do not ensure that these methods are practically effective—only empirical evidence can reveal whether they actually lead to meaningful learning. To validate the capacity of *BBoxER* as an LLM retrofitting method, we focus on verifiable domains, like math, where performance improvements are both measurable and interpretable. We experiment with budgets $b$ determined theoretically for guaranteed generalization on the test set and we empirically examine how performance changes as the budget exceeds these thresholds.

### 5.1 EXPERIMENTS

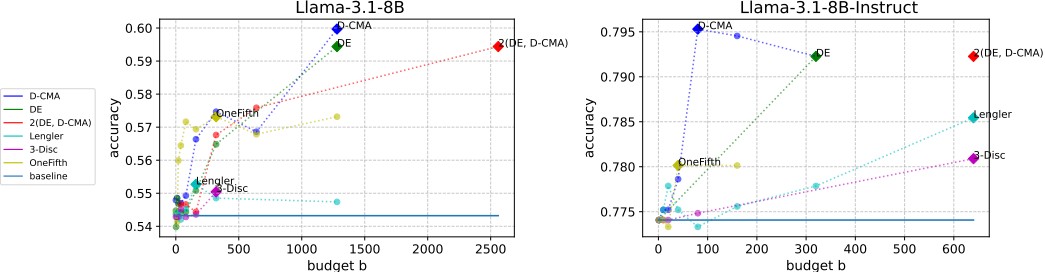

Figure 1: **Experiment 1 with Llama3.1-8B on GSM8K.** Accuracy (exact match) on GSM8k-test as a function of budget. **Left:** Base. **Right:** Instruct Model. D-CMA improves from 54.4% to 60.0% on base and from 77.4% to 79.5% on instruct. Notably, it improves even for budgets above those computed from our theory ($b > 150$).

**Experimental setup.** We deploy a family of BBO algorithms with finite branching factors, described in App. D, using the default parameters in Nevergrad. We conduct experiments on both base and instruct variants. Retrofitting the base models allows us to verify that learning indeed occurs, while targeting the instruct models demonstrates that our method can yield further improvements even on top of models already fine-tuned for reasoning tasks. All experimental details can be found in App. H.

**Datasets.** Most of our experiments deploy *BBoxER* on GSM8K (Cobbe et al., 2021), a dataset composed of short grade-school math problems, consisting of $7,473$ problems for training and $1,319$ for testing. We train on the train split and test on GSM8k-test (in distribution) with standard 8-shot methodology (unless otherwise stated), in addition to various other benchmark datasets (out of distribution): MATH500 (4-shots) (Hendrycks et al., 2021), Hellaswag (0-shots) (Zellers et al., 2019), GSM+ test_mini (8-shots) (Li et al., 2024a), ARC Easy (0-shots) and ARC Challenge (0-shots) (Clark et al., 2018), AMC23 (0-shots). All benchmarks are evaluated with exact match using greedy decoding. The theoretical bounds in Sec. 4 allow us to precisely determine the budget $b$ that guarantees

in-domain generalization (as a function of dataset size): for GSM8k-train the permissible budget is between 100-150 steps and approximately 3 times larger for Bet-and-run variants like 3-OneFifth.

**Choices for parameter updates.** A key design choice in our framework is choosing which parameters the BBO algorithm should modify. To explore this, we experiment with a couple of multiplicative update schemes, across different layers of the model: the outer (unembedding) layer (Experiment 1), $Q$ matrices of attention layers (Experiment 2) or all normalization layers (Experiment 3). We distinguish between *low-rank (LoRa)* - rank-1 in fact (Experiment 1), *broadcast* (Experiment 2) and *full* (Experiment 3) updates; *LoRa* multiplies the weight matrix by a low rank matrix, *broadcast* multiplies pointwise by a matrix with identical rows and *full* multiplies by an update matrix directly. For full details, refer to App. B. We optimize a small number of parameters, between $\approx 4K$ and $\approx 266K$, as BBO algorithms are known to scale poorly unless the problem's effective dimensionality is much lower (Nesterov & Spokoiny, 2017), which is likely the case here.

**Experiment 1 (rank-1, output layer updates).** In a first series of experiments on Llama3.1-8B (resp. Llama3.1-8B-Instruct) we test *BBoxER* performance on GSM8k-test, mostly with 1 seed (sometimes 2). Figure 1 illustrates the results: while D-CMA was our favorite choice, we ran other BBO algorithms to validate the robustness of the *BBoxER* approach.

**Experiment 2 (broadcast updates on attention layers).** In this setting, we again train a range of BBO algorithms on the GSM8K training set on Llama3.1-8B (Base) to explore out-of-distribution generalization capabilities of *BBoxER* with evaluations on SVAMP (Patel et al., 2021) and GSM+(Li et al., 2024a), in addition to GSM8k-test. We experiment with *broadcast* updates on $Q$-matrices of the 1st, 8th, or all attention layers. We have added bet-and-run variants which enjoy particularly strong generalization guarantees (see Sec. 4).

Tab. 1 shows the results for different settings. Each line without std.dev. ($\pm$) corresponds to one single run, and all runs use a budget $b = 150$ except otherwise specified; on the first attention layer, OneFifth was run with 3 seeds and D-CMA and CoLengler with 2 seeds each. Except for the "no few-shot" line, the default 8-shot evaluation is used on GSM8k. The results demonstrate the stability of *BBoxER* in terms of

Table 1: **Experiment 2**. Performance difference (after−before) *BBoxER* with Llama3.1-8B (Base) on GSM8k-train, evaluated on GSM8k-test (ID), (OOD) SVAMP and GSM+. Red: detrimental runs, $< -1\%$. Green: beneficial runs, $> 1\%$. Blue rows are bet-and-run cases.

| BBO algorithm | GSM8k | SVAMP | GSM+ |
|---|---|---|---|
| Base score | 56.93 | 76.6 | 51.34 |
| First attention layer, d=4096 | | | |
| OneFifth | $1.64\pm 1.28$ | $-1.0 \pm 1.08$ | $0.35 \pm 0.50$ |
| D-CMA | $0.49 \pm 0.19$ | $0.65 \pm 0.75$ | $0.00 \pm 0.02$ |
| COLengler | $-0.41 \pm 0.41$ | $-0.15 \pm 0.05$ | $-0.06 \pm 0.02$ |
| NgIohTuned | 1.28 | 0.90 | 0.18 |
| 3-OneFifth | 1.59 | -2.10 | -0.18 |
| All attention layers, d=131072 | | | |
| OneFifth | 1.13 | -0.80 | 0.19 |
| COLengler | 1.36 | -0.26 | 0.29 |
| 3-OneFifth | 2.27 | -0.10 | 1.01 |
| 3-OneFifth b300 | **7.12** | -0.90 | **4.99** |
| 3-OneFifth b450 | 5.61 | -0.20 | 4.85 |
| 3-OneFifth b600 | 5.61 | -0.20 | 4.85 |
| All attention layers, no few-shot, d=131072 | | | |
| Base score | 20.31 | 58.30 | 19.06 |
| 3-OneFifth | 1.97 | 3.30 | 0.04 |

learning capabilities, and a notable performance improvement of up to 7% in distribution (GSM8k test) and 5% in out-of-distribution transfer to GSM+. Additional results are reported in Tab. 6 in App. I.

**Experiment 3 (full update of normalization layers).** In this experiment, D-CMA is used to directly optimize all normalization layers of Llama3.1-8B(-Instruct) and Qwen2.5-3B-Instruct (no low-rank and no broadcast involved) on GSM8k. As in Experiment 2, we test both in distribution (GSM8k-test), and on various other benchmark datasets (out of distribution). We report the results averaged over 5 different runs with the standard deviation. The results (Tab. 3) indicate that running *BBoxER* improves in distribution performance and generally preserves it on very different tasks (e.g., Hellaswag). Additional experiments are described in App. I: BetAndRun results are reported in Tab. 7, and experiments using a larger training set can be found in Tab. 8 in App. I.

**Experiment 4 (Robustness to data extraction).** To assess the robustness of *BBoxER* against membership inference attacks (MIAs), we begin from the observation that most MIAs rely on the negative log-likelihood (NLL) of training samples as their primary signal to infer membership (Fu et al., 2023; Carlini et al., 2022; 2021) - see also App. H for an exposition. Accordingly, we propose to measure the average of absolute differences in negative log-likelihood (NLL) between the base and the updated model for each training sample.

Table 3: **Experiment 3 (full update, normalization layers)** Comparison of model performance of BBoxER run with D-CMA across various benchmarks, 5 seeds.

| Models | ID | OOD | | | | | |
|---|---|---|---|---|---|---|---|
| | GSM8k | GSM+ | MATH500 | ARC Easy | ARC Challenge | AMC23 | Hellaswag |
| Llama3.1-8B | 54.97 | 36.50 | 20.20 | 83.04 | 55.19 | 0.00 | 80.71 |
| Llama3.1-8B-BBoxER (b=150) | 55.24±0.92 | 37.28±0.48 | 19.72±0.68 | 83.05±0.12 | 54.97±0.12 | 4.50±2.92 | 80.67±0.05 |
| Llama3.1-8B-BBoxER (b=300) | 56.65±0.26 | 37.88±0.33 | 20.12±0.87 | 83.16±0.13 | 55.09±0.24 | 4.00±3.39 | 80.65±0.08 |
| Llama3.1-8B-Instruct | 77.26 | 54.37 | 37.60 | 79.62 | 55.45 | 22.50 | 79.90 |
| Llama3.1-8B-Instruct-BBoxER (b=150) | 77.79±0.41 | 55.11±0.15 | 37.16±0.53 | 79.16±0.32 | 55.48±0.17 | 25.00±3.54 | 79.98±0.05 |
| Llama3.1-8B-Instruct-BBoxER (b=300) | 78.57±0.41 | 55.33±0.53 | 37.56±0.51 | 79.64±0.12 | 55.61±0.15 | 21.00±3.00 | 79.94±0.04 |
| Qwen-2.5-3B-instruct | 79.98 | 62.29 | 41.00 | 72.39 | 47.38 | 32.50 | 74.94 |
| Qwen-2.5-3B-instruct-BBoxER (b=150) | 82.90±0.37 | 62.27±0.45 | 42.48±0.41 | 72.57±0.19 | 47.59±0.31 | 38.00±2.45 | 75.00±0.06 |
| Qwen-2.5-3B-instruct-BBoxER (b=300) | 83.55±0.36 | 61.94±0.64 | 41.32±0.95 | 72.62±0.17 | 47.76±0.22 | 36.50±3.00 | 75.02±0.09 |

We consider GSM8K as our training dataset, Llama-3.1-8B as our base model, and compute three metrics: Diff-full (NLL computed over the entire prompt), Diff-CoT-a (NLL computed over only the CoT and the answer), and the relative average exact match improvement on GSM8K-test from the base model. For a fair comparison, we evaluate *BBoxER* alongside standard fine-tuning and DP-AdamW, training all methods on GSM8K under different settings-varying budget, epochs, and parameter subsets (full details in App. H)—so that models are compared at similar utility. Our results, reported in Tab. 2, demonstrate that at matched test accuracy, the NLL shifts under *BBoxER* are orders of magnitude smaller than under fine-tuning or DP-AdamW. This provides empirical confirmation of our theoretical claim in Sec. 4.3: *BBoxER* is significantly more robust to extraction and membership inference attacks.

Table 2: **Experiment 4 - *BBoxER* is robust to MIA**. Comparison of *BBoxER*, SFT and DP-SFT in the setting of Experiment 3 with Llama3.1-8B-Instruct on GSM8K. Runs with comparable test accuracy are highlighted in the same color to compare robustness at similar accuracy. We see that both finetuning and DP-finetuning approaches lead to NLL metrics at least an order of magnitude larger than comparable *BBoxER* runs.

| Method | Diff-Full | Diff-CoT-a | Acc |
|---|---|---|---|
| ft-5epochs | 2.25e-01 | 5.06e-01 | +7.88 |
| ft-2epochs | 2.53e-01 | 3.99e-01 | +3.79 |
| ft-norm-5epochs | 6.00e-03 | 4.62e-02 | +0.45 |
| DP-AdamW-eps=8 | 8.66e-01 | 2.17e-01 | +2.05 |
| *BBoxER*-norm-b=150 | **7.19e-04** | **2.90e-03** | +0.15 |
| *BBoxER*-norm-b=300 | **1.01e-03** | **3.68e-03** | +1.97 |
| *BBoxER*-norm-b=1200 | **6.33e-03** | **1.17e-02** | +3.87 |

## 5.2 DISCUSSION OF EXPERIMENTAL RESULTS

**Robustness and statistical validation.** Our experiments demonstrate that *BBoxER* achieves statistically significant performance gains without overfitting across a range of BBO algorithms (see P-value validation in App. J), these gains are also observed on Instruct models. Consistent with theory, the Bet-and-Run variant 3-OneFifth further improves performance and allows for larger budgets as shown in experiment 2. More discussion is provided in App. I.

**Validation of transfer.** The compact optimization trace in *BBoxER* reduces the risk of overfitting to spurious patterns or idiosyncrasies in the training data. As a result, we may expect improved generalization and stronger performance on unseen datasets. In Experiments 2 and 3, indeed, we observe a improvement on GSM+ when evaluating our model trained exclusively on GSM8k. This positive transfer, in particular with 3-OneFifth, is especially noticeable when fine-tuning all the attention layers, highlighting the robustness of *BBoxER*. GSM+ is an interesting and challenging evaluation benchmark, as it modifies GSM8k questions to make them trickier by introducing various scenarios aimed at testing model robustness. In contrast, SVAMP presents simpler questions, typically involving only a single operation. As a result, improvements on SVAMP are more likely to reflect better numerical reasoning rather than enhanced problem comprehension.

**Computational cost.** A limitation of our approach is that each iteration requires a full pass over the dataset to generate a comparison result. However, *BBoxER* remains memory-efficient, as it relies solely on LLM inference without the need to compute and store gradients, or large optimizer states. This allows us to fit a large batch size during training. The overhead introduced by the internal state of the BBO algorithm is negligible compared to the computational cost of LLM inference.

## 6 CONCLUSION

This paper introduced *BBoxER* a principled post-training method that leverages comparison-based BBO algorithms for retrofitting. By compressing the training set into a comparison trace, measured in only a few bits, BBoxER offers strong theoretical guarantees on overfitting, robustness, and privacy. Despite this compression and the limited number of optimization iterations, BBoxER empirically demonstrates effective learning and transfer on GSM8k with contemporary LLMs, as well as empirical robustness to membership inference attacks. Theoretical insights are reinforced by experimental results, especially with Bet-and-Run algorithm variants. Importantly, this work is the first to bridge the LLM training landscape with the full BBO zoology. This connection, grounded in formal analysis, paves the way for safer, more robust, and adaptable language models, and opens the door to broader adoption of privacy-preserving training at scale, including completely private remote A/B testing or (post-)training directly on user satisfaction without any leakage of the underlying data.

**Reproducibility Statement** To enable independent re-implementation, we provide in the paper and Appendix: (i) The exact publicly available datasets and models used for training and evaluation (Sec. 5 and App. H) (ii) full hyperparameter details, $modifier$ functions, and training objective (Apps. B and H); (iii) number of few shots examples and inference settings for evaluation (Sec. 5); (iv) number of runs per result, with mean ± std when possible (Sec. 5); (v) open-source libraries used; and (vi) compute details (App. H).

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

## A EXTENDED RELATED WORK

**Generalization Bounds for LLMs.** Early generalization bounds, such as those derived from the VC dimension (Vapnik, 1995), covering numbers (van der Vaart & Wellner, 1996), and the fat-shattering dimension (Kearns & Schapire, 1990), become ineffective in the overparameterized regime of LLMs (Nagarajan & Kolter, 2019). PAC-Bayes-based methods have been proposed to derive bounds from model compressibility (Zhou et al., 2018; Lotfi et al., 2022; 2024), offering data-dependent generalization guarantees by leveraging informed priors over parameters to exploit the implicit preference of a network for simpler functions (Lotfi et al., 2022). This approach connects generalization to model compressibility: if a trained Transformer can be heavily compressed while retaining accuracy, it implies an effectively smaller hypothesis complexity. Indeed, some of the tightest known bounds come from compressing large networks. For instance, by quantizing parameters in a low-dimensional subspace. Such results echo Occam's razor, suggesting that big neural networks contain simpler sub-models that drive their generalization.

Another paradigm is algorithmic stability (Bousquet & Elisseeff, 2000), that instead analyzes how sensitive the learning algorithm is to perturbations in the training data. This makes it applicable to settings with infinite VC dimension. However, applying stability or norm-based analyses to huge LLMs often requires strong assumptions (e.g., tiny learning rates or bounded layer norms) and tends to yield loose bounds in practice. Hence, while these theoretical tools have advanced our understanding, for example, by explaining how a 100-billion-parameter Transformer might effectively behave like such a "simpler" model, extending them to practically useful guarantees is non-trivial. In particular, LLMs introduce extra complexities (e.g., autoregressive dependencies and unbounded loss values) that defy many standard assumptions. Li et al. (2023a) showed that the self-attention mechanism is uniformly stable under certain Lipschitz constraints, which allows a generalization bound for in-context learning with Transformers.

Our contribution shares this spirit, exploiting the comparison-based nature of optimization and the limited branching factor of black-box optimizers. By leveraging union-bound variants like the Bonferroni correction, we derive non-vacuous generalization guarantees.

**Data Poisoning.** Data poisoning alters training data to inject specific behaviors, either to degrade performance or to create targeted backdoors (Chen et al., 2017). This includes subtle manipulations—e.g., poisoning just $0.001\%$ of training data in medical applications (Alber et al., 2025)—that remain undetected by conventional metrics. Attacks have proven transferable across quantized and full-precision models (Egashira et al., 2024). Recent work shows that inserting only a few percent of malicious preference pairs in RLHF (Christiano et al., 2017) or DPO (Rafailov et al., 2023) data can dramatically shift model behavior (Baumgärtner et al., 2024; Wang et al., 2024; Yan et al., 2024; Zhong et al., 2023; Zeng et al., 2024; Zhao et al., 2025; Chen et al., 2017), such as increasing output verbosity or sentiment polarity. These changes are activated by specific triggers and survive quantization. Theoretical work (Kearns & Li, 1993; Steinhardt et al., 2017) confirms that even small adversarial corruptions can be highly effective, especially in overparameterized models. However, certified robustness against poisoning in LLMs remains an open problem. Compared to e.g., (Steinhardt et al., 2017), we provide an explicit algorithm with the desired stability assumptions.

**Privacy and Robustness.** LLMs are exposed to risks of privacy and data extraction attacks (Carlini et al., 2021; Zhang et al., 2024; Rando et al., 2025; Feng & Tramèr, 2024). DP-SGD (Abadi et al., 2016) introduces formal guarantees by adding Gaussian noise to clipped gradients, achieving $(\varepsilon, \delta)$-DP under strict accounting (Li et al., 2024b). Nevertheless, it suffers from increased compute, memory use, and degraded utility. Specific implementations like Book Keeping (Bu et al., 2023), and extensions like adaptive clipping and layer-wise noise scaling aim to mitigate this (Li et al., 2024b). PATE (Papernot et al., 2016) aggregates teacher votes to train a student model privately. Parameter-efficient strategies like DP-BiTFiT (Bu et al., 2024), soft prompts (Duan et al., 2023), and selective DP (Shi et al., 2022) offer better trade-offs. Our method avoids the need for noisy gradient updates by enforcing a compression bottleneck and comparison-based optimization, achieving $(\varepsilon = 0, \delta = 0)$ for user input prompts and outputs in scenarios like A/B testing, by targeting high-fidelity memorization pathways directly. This aligns with broader insights that data retention precision is a root cause of privacy and misuse (Lee & Yoon, 2025; Zheng et al., 2025). Yu et al. (2021) proposed a meta-framework for differentially private fine-tuning pretrained language models,

achieving state-of-the-art privacy-utility tradeoffs on standard NLP tasks. Their approach was inspired by the success of parameter-efficient fine-tuning methods, allowing for simpler and sparser algorithms while maintaining high utility and providing rigorous privacy guarantees. In this case, by leveraging DP-SGD with careful gradient clipping and noise addition, the framework ensures that the influence of any single data point on the model's parameters is bounded, thus protecting individual privacy. However, although the methods achieve strong performance, there is still a noticeable drop in accuracy compared to non-private fine-tuning, especially for smaller models or more complex tasks. Furthermore, the effectiveness of their DP fine-tuning approach is sensitive to hyperparameters such as learning rate, clipping norm, and noise multiplier, illustrating that careful fine-tuning is required to balance privacy and utility. Other works have focused on privacy-preserving strategies for instruction tuning and alignment of large language models. For instance, Yu et al. (2024) introduced a two-stage framework that addresses privacy risks not only during training, but also in the instruction collection phase. In this case, they first fine-tune a pretrained LLM using differentially private optimization (DP-Adam) to generate synthetic user instructions, and then apply a DP histogram-based distribution matching technique to resample instructions such that they resemble the distribution of the original private data. This approach ensures that both annotators and the final alignment pipeline never see raw user inputs, thereby preventing both direct exposure and downstream memorization. Applied to LLaMA (Touvron et al., 2023) and Phi-models (Li et al., 2023b), their approach achieved state-of-the-art performance in both supervised fine-tuning and RLHF settings, with utility comparable to non-private baselines. In particular, they highlight the importance of privacy-preserving data preparation in addition to model training.

**Post-training for Reasoning.** Reasoning tasks like GSM8k (Cobbe et al., 2021), SVAMP (Patel et al., 2021), MATH (Hendrycks et al., 2021), and GSM-Symbolic (Mirzadeh et al., 2025) highlight the limitations of standard LLM (pre)training. Post-training approaches for reasoning tasks are now commonplace in LLM training. Enhancements via chain-of-thought prompting (Wei et al., 2022b), complex reasoning extensions (Fu et al., 2022), and in-context learning (SU et al., 2023; Gupta et al., 2023) are effective but often brittle. Reinforcement learning approaches such as GRPO (Shao et al., 2024) allow fine-tuning using unsupervised signals, opening the door for real-time user-driven improvement (Peng & Risteski, 2022). LoRA (Hu et al., 2022) and robust tuning (Yen et al., 2025) enhance adaptability but are prone to hallucination (Ren & Sutherland, 2025) and exploitative behaviors (Qi et al., 2025; Zheng et al., 2025). Our approach combines low-rank adaptations with retrofitting techniques and avoids reward-based fine-tuning. It limits overfitting risks and prevents the amplification of memorized or poisoned content by emphasizing rank-based signal aggregation over token-level supervision. A somewhat complementary approach is self-improving (Sun et al., 2025). In such a case, a large language model tries to improve its performance by self-fine-tuning with synthetic data generated by itself. Nevertheless, such approaches have mostly focused on performance, and they are not risk-free from data poisoning or leaking private data.

## B EXAMPLES OF MODIFIERS

The *BBoxER* framework is described in Alg. 1. This algorithm has various hyperparameters, such as a budget, a black-box optimization algorithm, and a $modified(m_0, x)$ function which specifies how a vector $x$ proposed by the black-box optimization method is used to modify an initial model $m_0$. Here we present a few examples of such $modified(\dots)$ functions, involving broadcast, low-rank, or full updates.

### B.1 EXAMPLES OF MODIFIERS FOR ONE LAYER

In the present section $\times$ refers to pointwise multiplication and @ refers to matrix multiplication.

**Full Layer Update.** A simple example of a multiplicative modifier is the following:

1: Define $model = modified(m_0, x)$
2: $model = copy(m_0)$
3: $model.output\_matrix = model.output\_matrix \times \exp(0.01 \times x)$
4: Return $model$

In this example, we modify the normalization layer of the model, using a matrix $x$.

Other update formulas include $+0.01 \times x$, $\times (Id + 0.01 \times x)$, and other constants than $0.01$. However, in most cases, we will use vectors $x$ rather than matrices, as explained in low rank and broadcast.

**Low Rank Update.** When updating entire weight matrices, we prefer to work with low-rank (**LoRA**) updates (rank 1 in fact), defined as follows, when $model.output\_matrix$ has shape $(hidden\_dim, vocab\_size)$ and $x$ has dimension $hidden\_dim + vocab\_size$:

1: Define $model = modified(m_0, x)$
2: $model = copy(m_0)$
3: $x_1, x_2 = split(x)\{x_1$ has shape $(hidden\_dim, 1)$ and $x_2$ has shape $(1, vocab\_size)\}$
4: $model.output\_matrix = model.output\_matrix \times \exp(0.01 \times x_1 @ x_2)$
5: Return $model$

**Broadcast Update.** In some cases, we use **broadcast** rather than low-rank updates. $modified(m_0, x)$ is then defined as follows, where $x$ as a row vector is repeated $M$ times to fit the shape of the matrix we want to update. In Sec. 5 $x$ has dimension $hidden\_dim$ and $M = hidden\_dim$:

1: Define $model = modified(m_0, x)$
2: $model = copy(m_0)$
3: $x_1 = (1, \ldots, 1)$ $\{x_1$ has shape $(1, hidden\_dim)\}$
4: $model.output\_matrix = model.output\_matrix \times \exp(0.01 \times x_1^t @ x)$
5: Return $model$

In App. I, $x$ has dimension $vocab\_size$ when broadcast updates are applied to the output layer.

### B.2 MODIFYING SEVERAL LAYERS

In the examples above, we consider updates of one layer only. However, this can be naturally extended to updates of $l$ layers. Then, $x$ is split into $x_1, \ldots, x_l$, and each $x_i$ is used for updating a specific layer, as in App. B.1.

In our experiments in Sec. 5 we performed the following updates:

- Optimizing the output layer with the rank 1 Low Rank Update. (Experiment 1)
- Optimizing all $Q$ matrices of all attention layers with the Broadcast Update. (Experiment 2)
- Optimizing all normalization layers with the Full Layer Update. (Experiment 3)

For all these experiments the update constant is $0.01$. A few additional experiments in App. I use a different constant.

## C THEORETICAL BACKGROUND: A BRIEF OVERVIEW OF STATISTICAL LEARNING THEORY

### C.1 NON ASYMPTOTIC LARGE DEVIATION BOUNDS FROM THE 60S: HOEFFDING, BENNETT, BERNSTEIN

Large deviation bounds refer to bounds on the difference between an expectation and an empirical average. They exist independently of machine learning (contrary to overfitting bounds, below) and are used for polls, Monte Carlo integration, and others.

Indeed, for a dataset size $s$, consider $\delta_{1,\epsilon}$ the risk against the randomness $\omega_D$ w.r.t. dataset $D$ of an $\epsilon$-divergence between the empirical risk and the risk in generalization for a given parametrization $x$, i.e.,

$$\forall (x, \epsilon, s), P_{\omega_D}\left(|\hat{L}(x) - L(x)| > \epsilon\right) \leq \delta_{1,\epsilon}. \tag{A.1}$$

$\delta_{1,\epsilon}$ refers here to the risk of deviation $\epsilon$ for one (hence the subscript 1) model: when $\hat{L}$ is computed by average on a sample, we can invoke $\delta_{1,\epsilon}$ by Hoeffding, Chernoff or Bernstein bounds (Devroye et al., 1996). By Hoeffding's bound (Hoeffding, 1963), if individual losses are in $[0, 1]$ we get

$$\delta_{1,\epsilon} = 2 \exp(-2s\epsilon^2). \tag{A.2}$$

In case we have a prior on a bound $\sigma^2$ on the variance of a randomly drawn individual loss, we can use the bound of Bennett and Bernstein (Bennett, 1962):

$$\delta_{1,\epsilon} = 2 \exp\left(-s\epsilon h_1\left(\frac{\epsilon}{\sigma^2}\right)\right).$$

where $h_1(\lambda) = (1 + \frac{1}{\lambda})\ln(1 + \lambda) - 1$.

### C.2 PROBABLY APPROXIMATELY CORRECT (PAC) LEARNING: A BRIEF OVERVIEW

We propose a point of view on optimization algorithms based on their risk of overfitting in the context of huge VC-dimensions and covering numbers, deriving bounds that are not covered by the studies on the rates of convergence. These bounds were frequently developed in the '90s or '00s:

- App. C.2.1 presents the Bonferroni correction: whereas "non-uniform" large deviation inequalities estimate the difference between an empirical error and a generalization error for a single predefined classifier (for example when testing a single model on a test set), the Bonferroni correction provides bounds applicable uniformly on a finite set of classifiers (e.g., for selecting, with a validation set, a classifier in a list of classifiers obtained on the training set).

- App. C.2.2 generalizes the Bonferroni correction to infinite families of classifiers: VC-dimension bounds, and others, are the most well-known bounds. They are based on limiting the "capacity" of a space of function, even when it is infinite.

- App. C.2.3 presents other bounds for infinite families of functions, taking into account the optimization algorithm used for selecting a classifier: the idea is that even if the space of functions is large, we can get bounds if the optimization algorithm satisfies some stability assumptions. This last part of the state-of-the-art is the closest to our work: we get non-trivial bounds on deep learning models, independently of the possibly huge number of parameters.

#### C.2.1 BOUNDS ON THE GENERALIZATION LOSS BASED ON BONFERRONI

The Bonferroni bound (Bonferroni, 1936; Dunn, 1961) is a direct application to statistics of the union bound:

$$P(A_1 \text{ or } A_2 \text{ or } \ldots \text{ or } A_k) \leq \sum_{i=1}^{k} P(A_i) \tag{A.3}$$

The analysis below extends (Videau et al., 2024c;b), using the branching factor analysis in (Fournier & Teytaud, 2011). $L_F(x)$ refers to the error in generalization for a probability distribution $F$ and a model $x$. $\hat{L}_D(x)$ (frequently denoted $\hat{L}(x)$ in the literature) refers to the random variable corresponding to the empirical error on a dataset $D$, of cardinality $s$, randomly independently drawn from $F$. Whereas gradient-based optimization does not lead to any bound on the generalization loss, such bounds can be obtained when using comparison-based optimization (i.e., a subset of black-box optimization), as detailed below. These bounds suggest that the risk of overfitting is lower than for gradient-based methods. Our experimental results suggest that this interpretation is correct.

Let us see the generalization of large deviation inequalities seen in App. C.1.

If we use the empirical error for picking up the best $\hat{x}$, in a list of $b$ models $x_1, \ldots, x_b$ (e.g., $x_1, \ldots, x_b$ are chosen by random search or low-discrepancy search or choice by the user, which leads to $N(\omega, a, b) = b$), the risk of deviation $\epsilon$ for that $\hat{x}$ is at most $b \cdot \delta_{1,\epsilon}$ (union bound, also known as Bonferroni correction (Bonferroni, 1936)) instead of $\delta_{1,\epsilon}$: $P_{\omega_d}(|\hat{L}(\hat{x}) - L(\hat{x})| > \epsilon) \leq \delta_\epsilon = b\delta_{1,\epsilon}$.

Thm 1 extends this to more sophisticated cases, with $N(\omega, a, b) = \prod_{i=1}^{b} k_i(\omega, a, b) >> b$ (in many cases, exponential in $b$ e.g. $N(\omega, a, b) \leq 2^b$ and $k_i \leq 2$) instead of $N(\omega, a, b) = b$. In spite of being larger, this bound, exponential in $b$, is tighter than bounds based on capacity.

### C.2.2 BOUNDS ON THE GENERALIZATION LOSS BASED ON CAPACITY: STATISTICAL LEARNING THEORY FROM THE 90S

Vapnik (1995) is a pioneering work on bounds on the generalization loss based on VC-dimension. Many other capacity measures have been defined, such as covering numbers (van der Vaart & Wellner, 1996) and the fat-shattering dimension (Kearns & Schapire, 1990). The most classical bound on the generalization loss is probably the VC-bound in the case of classification:

$$\mathbb{E}[L(\hat{x})] \leq \inf_x L(x) + 16\sqrt{\frac{VC\log(s) + 4}{2s}}$$

(assuming that $\hat{x}$ minimizes $\widehat{L}(\widehat{x})$). When using shattering coefficients instead of VC-dimension, one can read:

$$\mathbb{E}[L(\hat{x})] \leq \inf_x L(x) + 16\sqrt{\frac{\log(8e \times Shattering(n))}{2s}}.$$

When it is possible to reach zero loss, then we get

$$P(L(\hat{x}) > \epsilon) \leq 2 \times Shattering(2n)2^{-s\epsilon/2}.$$

Both the Shattering coefficients and the VC-dimension depend on the class of functions. Extensions exist for regression and other cases, and other capacity measures exist. However, capacity bounds are not always relevant for vast models such as deep networks.

### C.2.3 ALGORITHMIC STABILITY: BOUNDS BASED ON PROPERTIES OF ALGORITHMS IN THE 2000S

Algorithmic Stability (Bousquet & Elisseeff, 2000) is a radically different approach, based on some algorithm properties rather than on the capacity of the underlying family of functions. Because we also use particularities of the learning algorithm to derive generalization bounds, our approach is close to that of Algorithmic Stability.

Following (Bousquet & Elisseeff, 2000), we write $l(x, z)$ the loss of a model parametrized by $x$ on an example $z$, so that $L(x) = \mathbb{E}_z l(x, z)$. An algorithm which outputs a model parametrized by $\hat{x}(D)$ on the training data $D$ has uniform stability $\beta$ if $\sup_{D,z} |l(\hat{x}(D), z) - l_{\neg i}(\hat{x}(D), z)| \leq \beta$ for all $i$, where $l_{\neg i}$ is the loss of the model learned from a training set obtained from $D$ by removing the $i^{th}$ example. An algorithm with uniform stability $\beta$ and some other technical assumptions (including $0 \leq l \leq 1$) has the following property:

$$\forall \delta > 0, P\left(L(\hat{x}(D)) > \hat{L}(\hat{x}(D)) + \sqrt{\frac{1 + 12s\beta}{2s\delta}}\right) < \delta.$$

## D BLACK-BOX OPTIMIZATION ALGORITHMS

The *BBoxER* algorithmic framework, as illustrated in Figure 2, embeds an algorithm from the family of block-box optimization algorithms, which we now describe in detail.

### D.1 EVOLUTIONARY ALGORITHMS

For the reasons given in Sec. 3, only Evolutionary Algorithms (EAs) have been used in the present work, as they are (at least for those uses in the present document) comparison-based, and hence de facto comply with the finite branching factor condition, which is mandatory for the theoretical results of Sec. 4 to hold. App. D.2 discusses this choice, and its possible relaxation, while the current section introduces EAs more thoroughly, and details all the comparison-based algorithms used in the experiments of Sec. 5.

Evolutionary algorithms (A. E. Eiben and, 2015) are stochastic optimization algorithms that crudely mimic the Darwinian evolution of a "population" of $\mu$ "individual" (i.e., points of the search space) based on random variations and "survival of the fittest" selection: the $\mu$ "parents" generate $\lambda$

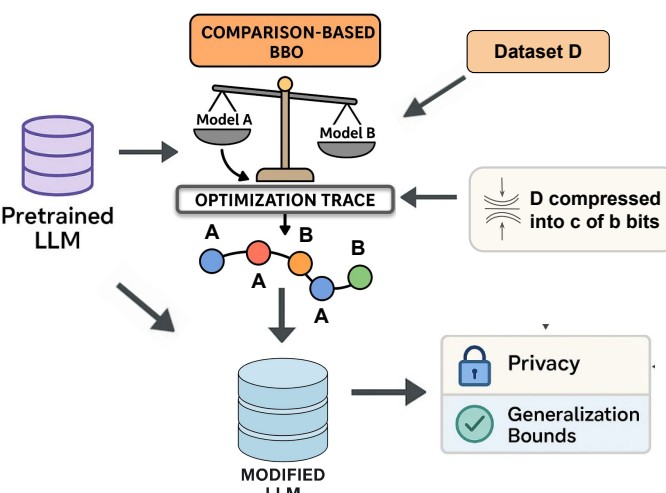

Figure 2: **Schematic view of *BBoxER* Alg. 1.** Our approach compresses the data relative to the current model, leading to a list $c$ of $b$ bits (Eq. 3, $b$ is the total budget). The final model depends only on those $b$ bits, giving perfect control over the number of bits transferred from the dataset to the model. In *BBoxER*, budgets $b$ are typically between 60 and 1500 bits, i.e., very small compared to the dataset size. The "Retrofitting compressor" refers to the part of Alg. 1 which outputs the list $c$ of outputs of the $Compare(\dots)$ function: $c$ represents a compression of the dataset $D$ conditional on the initial model. "Retrofitting modifier" is the part of Alg. 1 which outputs the final model, given the list $c$ and the initial model.

"offspring" by random operations; the selection process then retains $\mu$ individuals to build the next population; in $(\mu + \lambda)$-EAs, these are the fittest of the $\mu$ parents plus the $\lambda$ offspring, while $(\mu, \lambda)$-EAs select the best of the $\lambda$ offspring only. EAs can search any search space on which random variation operators can be defined. These EAs are **comparison-based**, i.e., only comparisons of the values of the objective are used (to deterministically choose the best ones). But stochastic selection processes can also be used in general EAs.

When it comes to continuous optimization (the variables are continuous), among the most popular EAs are Evolution Strategies (ESs) (Beyer & Schwefel, 2002), Differential Evolution (DE) (Storn & Price, 1997), and Particle Swarm Optimization (PSO) (Kennedy & Eberhart, 1995).

Evolution Strategies (Rechenberg, 1973; Schwefel, 1977; Beyer & Schwefel, 2002) are specific EAs for continuous optimization, using normal "mutations", i.e., sampling some Gaussian distribution, generally centered on the parent, to generate the offspring. The issue is then to adjust the variance and the covariance matrix of this normal mutation.

The first adaptive ES uses the so-called "one-fifth success rule" (Schumer & Steiglitz, 1968; Rechenberg, 1973), established by theoretical analyses on simple linear and quadratic objective functions, on which the optimal behavior is reached for a success rate (offspring better than parent) of approximately 0.2. This is available in Nevergrad as the *(1+1)-ES with 1/5th rule*, named in this paper **OneFifth**, whose pseudo-code is given in Alg. 2-left.

**CMA-ES** (Hansen & Ostermeier, 1996; 2003; Hansen, 2023) iteratively updates a multivariate Gaussian random variable, initialized at the identity and parametrized by a covariance matrix and its variance (the so-called *step-size*). Each update is based on the correlations observed in successful mutations. This allows CMA-ES to adapt its step-size and search directions (the covariance matrix) to the problem landscape. Despite its $o(d^2)$ computational cost and memory usage, due to the handling of the full covariance matrix, it is widely regarded as a powerful and reliable optimizer.
**D-CMA** (Ros & Hansen, 2008) is the most popular of the several "tricks" that have been proposed to cope with this complexity. D-CMA uses a diagonal covariance matrix rather than a full positive

---

**Algorithm 2** The two typical (1+1)-EAs.

**Left**: all variables are added some Gaussian noise in the $(1 + 1)$-ES, but the step-size of the multivariate Gaussian mutation is modified according to the success rate.

**Right**: each variable is replaced with a given probability by a value sampled from a fixed standard normal Gaussian in all discrete-inspired (1+1)-EA.

| **Adaptive One-fifth $(1 + 1)$-ES** | **Generic Discrete-inspired $(1 + 1)$-EA** |
|---|---|
| **Require:** initial point $x_0 \in \mathbb{R}^d$; budget $b \in \mathbb{N}$ | **Require:** initial point $x_0 \in \mathbb{R}^d$; budget $b \in \mathbb{N}$ |
| **Require:** step-size $\sigma > 0$ | 1: $x = x_0$ |
| 1: $x = x_0$ | 2: **for** $i \in \{1, \ldots b\}$ **do** |
| 2: **for** $i \in \{1, \ldots b\}$ **do** | 3: $\quad x' = x$ |
| 3: $\quad x' = x + \sigma \mathcal{N}(0, 1)$ | 4: $\quad$ **while** $x' = x$ **do** |
| 4: $\quad$ **if** $x'$ better than $x$ **then** | 5: $\quad\quad x' = x$ |
| 5: $\quad\quad x = x'$ | 6: $\quad\quad$ **for** $j \in \{1, \ldots, d\}$ **do** |
| 6: $\quad\quad \sigma = 2\sigma$ | 7: $\quad\quad\quad$ With probability $p(i, b, d)$ $\quad x'_j = \mathcal{N}(0, 1)$ |
| 7: $\quad$ **else** | 8: $\quad\quad$ **end for** |
| 8: $\quad\quad \sigma = 2^{-\frac{1}{4}}\sigma$ | 9: $\quad$ **end while** |
| 9: $\quad$ **end if** | 10: $\quad$ **if** $x'$ better than $x$ **then** |
| 10: **end for** | 11: $\quad\quad x = x'$ |
| | 12: $\quad$ **end if** |
| | 13: **end for** |

definite matrix. Its efficiency, hence, depends on the degree of correlation between the variables, aka the *separability* of the objective function.

When handling a full population ($\mu > 1$), both CMA-ES and D-CMA apply the adaptive Gaussian mutation to a linear recombination of the $\mu$ individuals. The historical CMA-ES used the mean (all weights of the linear recombination are $1/\mu$), whereas the state-of-the-art CMA-ES uses weights that decrease with the rank of each individual. In the latter case, more information is used, and the branching factor increases (see App. E.5).

The simplest ESs (or EAs) are the (1+1)-EAs: the single current individual is modified, and the best of both becomes the new current one. While the already-mentioned **OneFifth** was the first historical successful adaptive EA (see again Alg. 2-Left), another family of (1+1)-EAs is directly derived from the historical GAs (Genetic Algorithms), that optimize bitstrings: the mutation typically flips each bit with a given probability. In the continuous case, the bitflip is turned into a "Gaussian flip": each value is replaced by a value sampled from a standard normal mutation ($\mathcal{N}(0, 1)$ in Alg. 2). Several variants of such "discrete-inspired" algorithms exist in Nevergrad, depending on the probability of applying this Gaussian flip to each variable. The **Discrete** algorithm exactly mimics the standard setting in GAs by adopting a fixed $1/d$ probability; the **Lengler** (Doerr et al., 2019; Einarsson et al., 2019) algorithm uses a gradually decreasing schedule. The **Uniform-Disc** (Dang & Lehre, 2016) algorithm uses a probability that is uniformly sampled in $]0, 1]$, and the **FastGA** (Doerr et al., 2017) uses an $\alpha/d$ probability where $\alpha$ is sampled from a power-law distribution.

Additionally, **COLengler** is a variant of Lengler that also performs, after mutation, a coordinate-wise crossover *à la* differential evolution.

Table 4: Bound on the average branching factor for various algorithms (see App. E.4, App. E.5, App. E.6).

| Algorithm | $\left(\prod_{i=1}^{b} k_i\right)^{1/b} \leq$ |
|---|---|
| **$(1 + 1)$ algorithms** | |
| OneFifth | 2 |
| Lengler | 2 |
| FastGA | 2 |
| Portfolio | 2 |
| **Bet-And-Run algorithms** | |
| 3-Disc | $\left(3 \cdot 2^{b/3}\right)^{1/b} < 2$ |
| 3-OneFifth | $\left(3 \cdot 2^{b/3}\right)^{1/b} < 2$ |
| **Population-based algorithms** | |
| DE, PSO | 2 |
| DE, PSO with curr-to-best | 3 |
| CMA-ES/D-CMA | 2 |
| CMA-ES/D-CMA with distinct weights | $2 \times \sqrt{\mu}$ |

**DE** (Differential Evolution) (Storn & Price, 1997) evolves a population of individuals: each individual is mutated using (i) differences between good points (for taking into account the key directions) and, in its "**Curr-to-best**" variants (ii) typical differences between the population and the best point so far, before a coordinate-wise crossover is applied. In spite of its very low computational cost, DE is known as a very efficient algorithm.

**PSO** (Particle Swarm Optimization) (Kennedy & Eberhart, 1995) uses a population of moving individuals ("particles"), each with an associated velocity. At each iteration, the velocity of each particle is biased towards the best value observed so far by this particle, and towards the best value over the complete population of particles, and the particle moves according to its new velocity.

Furthermore, Nevergrad also includes some *ensemble approaches*.

**NgIohTuned** is a *wizard*. It chooses one algorithm from its portfolio of BBO algorithms, based on some characteristics of the problem, and following a hand-made rule. Portfolios and wizards have been first proposed for SAT solving (Xu et al., 2008), and later ported to black-box optimization (Awad et al., 2020; Meunier et al., 2021).

**Bet-And-Run** algorithms, for budget $b$, consists in running $k$ algorithms with budget $\alpha b/k$, picking up the one reaching the best objective value and running it for the remaining $(1 - \alpha)b$ budget; **3-OneFifth** is an instance of Bet-And-Run with $\alpha = 0.5$ and three OneFifth in parallel; with $\alpha = 1$, **3-Disc** runs three Discrete in parallel and **2-DE/D-CMA** runs DE and D-CMA.

These algorithms were chosen because of their finite branching factor: Tab. 4 lists the branching factors of all cited algorithms. The branching factor is 2 for all (1+1) algorithms. The derivation of the other values is detailed in App. E.

**Discussion.** The choice of a BBO algorithm for a given problem is a long-lasting problem, and can of course be left to the Nevergrad wizard NgIohTuned. However, some characteristics of the algorithm introduced above are worth highlighting here, and might in turn give some information about the problems addressed with *BBoxER* based on its experimental results.

The adaptive approaches are now well mastered: CMA-ES offers the most sophisticated update mechanism. It is competitive in terms of performance with the best algorithms of Mathematical Programming, and has recently been proven to converge linearly (Gissler, 2024) However, it has a high computational complexity with respect to dimension $d$ and hence poorly scales up. Among several variants, D-CMA offers a very good compromise. But the simplest adaptive mechanism is that of OneFifth, which is very robust and often very efficient. Note, however, that its adaptation mechanism requires a number of iterations to take place, and was, for example, discarded in the OpenAI ES, one of the rare examples of large-scale BBO in complete training of Deep Neural Networks (Salimans et al., 2017).

Because D-CMA only adapts a diagonal covariance matrix, one cannot expect that it effectively handles non-separable objectives. Less intuitively, this is also true for PSO, and, to a lesser extent, for DE unless its crossover rate is small (Auger et al., 2009). On the other hand, their adaptive mechanism (implicit for DE) makes CMA-ES, D-CMA and (some variants of) DE very efficient in dealing with randomly rotated ill-conditioned problems, which is not the case for PSO (Auger et al., 2009).

Regarding the discrete-inspired (1+1)-EAs, Discrete, Uniform-Disc, Lengler, and FastGA, they are based on proved convergence proofs for known benchmark functions in the bitstring case, but their efficiency in the continuous setting is more surprising, and would deserve deep mathematical analyses.

Last but not least, Bet-And-Run algorithms, including 3-OneFifth, 3-Disc, and 2-DE/D-CMA, are robust, as proved in App. E.4 and demonstrated in the experimental results (Sec. 5 and App. I. However, they still require the choice of the embedded algorithms.

### D.2 POSSIBLE EXTENSIONS

**Code and Package.** As said, all experiments in this work have been run on the Nevergrad open source platform (Rapin & Teytaud, 2018), which encompasses many BBO algorithms, along with specialized wizards that select among them based on problem-specific properties. Nevergrad also includes algorithms that are black-box optimizers without being comparison-based, such as (Powell, 1994; 1964). Since these operate with a finite number of bits per loss function, they might also fall under the theoretical framework of Sec. 4, though this is left for future work.

Extension to multi-objective algorithms is another promising direction, for instance by optimizing different criteria (e.g., pass@k and accuracy) or by considering performance across several distinct benchmarks as the training objective, rather than averaging them.

**Beyond Comparison-based Algorithms** An important assumption (Line 5) is that the only thing that depends on the dataset $D$ is $choice_1, \ldots, choice_b$, and each $choice_i$ has at most $k_i$ possible values ($k_i := alg.numCalls()$ is called the branching factor (Fournier & Teytaud, 2011)). This is unusual in deep learning (DL), as in DL *compare* is replaced by some function involving gradients (even Reinforce (Williams, 1992) or GRPO (Shao et al., 2024) return a gradient update), and $choice_i$

is replaced by a gradient, or, for reinforcement learning such as GRPO (Shao et al., 2024), the outputs and their rewards.

Algorithms that do not return any gradient are termed black-box optimization algorithms. Among black-box optimization algorithms, some methods are comparison-based: they have finite $k_i$. These comparison-based algorithms are typically classified in three families of algorithms: (i) random search and derandomized variants (McKay et al., 1979; Niederreiter, 1992), (ii) direct search methods (Roberts & Royer, 2023), and (iii) evolutionary computation (Beyer, 2001). Note that GRPO (Shao et al., 2024) is sometimes considered as gradient-free in the sense that it does not require the gradient of the loss function; however, it does require other detailed information about the outputs of models, so that it is not black-box in the same sense (and a fortiori not comparison-based), and the present results do not apply to GRPO or REINFORCE (Williams, 1992) if we assume that they work on infinite-precision floats.

**Taking into account the Finite Float Precision.** Because our results are mainly useful when $k_i$ is small, this work focused on comparison-based algorithms. However, all results of Sec. 4 are true for large values of $k_i$. Let us check $k_i$ in the case of some classical methods for post-training. In *BBoxER*, the information flow from the data to the model is limited to roughly one bit per iteration, leading to at most $2^b$ possible outputs for a budget of $b$ steps. This limited capacity directly reflects the compression properties showcased in Equation (3) and Figure 2. In contrast, supervised fine-tuning updates the model using one gradient per mini-batch. For an 8B model with 16-bit precision, each mini-batch carries about 256B bits: a few mini-batches are sufficient to encode the entire content of Wikipedia. Reinforcement learning methods, such as GRPO, typically rely on model outputs and reward signals that would allow them to encode the full dataset within a single training epoch.

# E  OVERFITTING BOUNDS AND THEIR GENERALIZATIONS

In this Section we provide the theoretical generalization bounds for the deterministic (App. E.1) and randomized cases (App. E.2), followed by (App. E.3) strengthened bounds for the special cases of $(1, \lambda)$ and $(1 + \lambda)$ BBO algorithms as described in App. D. We proceed by making our bounds explicit for various families of algorithms, based on Fournier & Teytaud (2011); in App. E.4 we obtain strengthened bounds for Bet-and-Run combinations like 3-OneFifth, MultiDisc, or 2-DE/D-CMA (deployed in some of our experiments in Sec. 5), App. E.5 lays out bounds for population-based evolution strategies (i.e., $(\mu + \lambda)$ and $(\mu, \lambda)$ with $\mu > 1$) like CMA-ES and D-CMA, and App. E.6 deals with differential evolution and particle swarm optimization. The key to our results is the branching factor $k_i$, i.e., the number of distinct behaviors of the algorithm that can arise in an iteration. Finally, App. E.7 derives explicit bounds as a function of the dataset size.

## E.1  BOUNDS FOR THE DETERMINISTIC ALGORITHM: THEOREM 1

**Theorem 1** (Deterministic case). *Let $a$ be an algorithm, $b$ a budget, $s$ the dataset size, and $\epsilon > 0$. Assume $N(\omega, a, b)$ is finite and that for some $\delta_{1,\epsilon}$, $\forall x$, $P_{\omega_D}\left(|\widehat{L}(x) - L(x)| > \epsilon\right) \leq \delta_{1,\epsilon}$. Then*

$$P_{\omega_D}\left(|\widehat{L}(\widehat{x}) - L(\widehat{x})| \geq \epsilon\right) \leq N(\omega, a, b) \cdot \delta_{1,\epsilon}. \tag{A.4}$$

$$P_{\omega_D}\left(\sup_{1 \leq i \leq b} |\widehat{L}(x_i) - L(x_i)| \geq \epsilon\right) \leq b \cdot N(\omega, a, b) \cdot \delta_{1,\epsilon}. \tag{A.5}$$

*Proof.* For Equation (A.4): apply the Bonferroni bound (Equation (A.3)) to the set of events

$$\{|\hat{L}(FinalModel_{\omega,D,a,b}) - L(FinalModel_{\omega,D,a,b})| > \epsilon; D \in \mathcal{D}\}.$$

This set has cardinality at most $N(\omega, a, b)$.

For Equation (A.5), apply the Bonferroni bound to the set of events

$$\{|\hat{L}(model_{\omega,D,a,b,i}) - L(model_{\omega,D,a,b,i})| > \epsilon; i \in \{1, \ldots, b\}, D \in \mathcal{D}\},$$

where $model_{\omega,D,a,b,i}$ is $m_i$ when *BBoxER* is applied with seed $\omega$, algorithm $a$, budget $b$, dataset $D$. This set has cardinality $b \cdot N(\omega, a, b)$. □

### E.2 GENERALIZATION BOUNDS FOR RANDOMIZED *BBoxER*

Thm 1 applies to deterministic algorithms, where the only source of randomness is the sampling of the dataset $D$, determined by the seed $\omega_D$. In this case, the deviation risk $\delta_{1,\epsilon}$ is computed over that randomness alone. When the optimization algorithm itself is randomized (i.e., $\omega$ is a random seed), the deviation risk $\delta_{\epsilon,\omega}$ also depends on $\omega$. Averaging over $\omega$ yields the total deviation risk:

$$\delta_\epsilon = \mathbb{E}_\omega \delta_{\epsilon,\omega}.$$

Since the bound of Thm 1 holds uniformly for any fixed $\omega$, it extends naturally to the randomized setting, in which $\widehat{x}$ depends on both $\omega$ and $\omega_D$:

> **Theorem 3** (Stochastic case). *Consider a randomized seed $\omega$ (i.e. $\omega$ is a random variable), an algorithm $a$ and a budget $b \in \mathbb{N}$. Then*
>
> $$P_{\omega,\omega_D}(|\hat{L}(\widehat{x}) - L(\widehat{x})| \geq \epsilon)) \leq \left( \sup_{\substack{\omega \ constant}} N(\omega, a, b) \right) \cdot \delta_{1,\epsilon}.$$

Here, the probability is taken over both $\omega$ (which determines the optimization trajectory and hence the final model $\widehat{x}$) and $\omega_D$ (which governs dataset sampling). The supremum is over a constant $\omega$, and $\widehat{x}$ can be built with a random variable $\omega$.

**Application to Random Search.** In this case, the $b$ candidate models are sampled independently of the data. Since each choice is fixed a priori, the number of internal states satisfies $N(\omega, \texttt{random-search}, b) \leq b$. $N(\omega, a, b)$ is derived from $k_b = b$ (we choose between the $b$ models $x_1, \ldots, x_n$ at the very end) and $k_i = 1$ for other values of $i$. Apply the Bonferroni correction to the $N(\omega, a, b)$ possible values of $\widehat{x}$ leads to

$$P_{\omega_D}(|\widehat{L}(\widehat{x}) - L(\widehat{x})| \geq \epsilon) \leq b \cdot \delta_{1,\epsilon}. \tag{A.6}$$

This also applies to quasi-random strategies and design of experiments, where the sampling of hyperparameters does not depend on their empirical performance. In these cases, we observe that $b$ can be exponential in $s$ without overfitting.

### E.3 COMPARISON-BASED STRATEGIES: THE $(1, \lambda)$ AND $(1 + \lambda)$ CASES.

In contrast to random search, where all candidate parametrizations are chosen independently of their evaluation, comparison-based algorithms build candidates iteratively based on performance. If $\lambda$ candidates are evaluated at each step and the best is retained, then the number of internal states after one iteration is $\lambda$; after two iterations, $\lambda^2$; and after $n = b/\lambda$ iterations, at most $\lambda^n$. The total risk is, therefore, bounded by $\lambda^n \cdot \delta_{1,\epsilon}$, where $n$ is the number of iterations with $k_i > 1$ (with $k_i = \lambda$ in such iterations); see (Fournier & Teytaud, 2011).

In the elitist variant $(1+\lambda)$, the best model is selected among the current best and the $\lambda$ new candidates, so $k_i \in \{1, \lambda+1\}$. This leads to a total number of internal states bounded by $(\lambda + 1)^{(b-1)/\lambda}$. Simply apply Thm 3 in the case

- $k_i = \lambda$ if $i + 1 \mod \lambda = 0$ (every $\lambda^{th}$ iterations we pick up the best of the previous $\lambda$ models).
- $k_i = 1$ otherwise.

and the corresponding generalization bound becomes:

$$P(|\widehat{L}(\widehat{x}) - L(\widehat{x})| \geq \epsilon) \leq (\lambda + 1)^{(b-1)/\lambda} \cdot \delta_{1,\epsilon}. \tag{A.7}$$

For example, this applies to the $(1 + 1)$ evolution strategy with the one-fifth success rule or the Lengler variant (Einarsson et al., 2019), both corresponding to $k_i = 2$.

### E.4 GENERALIZATION BOUNDS FOR BET-AND-RUN COMBINATIONS

Consider the example in Equation (A.7). Additionally, if we perform $k > 1$ independent optimization runs, and if the budget $b$ is divided by $k$, then the Bonferroni correction leads to a risk

$$\underbrace{k}_{\text{Bonferroni correction}} \cdot \underbrace{\lambda^{(b/(\lambda \cdot k))}}_{\text{Branching factor per run}} \cdot \underbrace{\delta_{1,\epsilon}}_{\text{Risk for a single model}} \tag{A.8}$$

for the resulting parametrization. According to these bounds, increasing the number $k$ of independent runs for a given total budget decreases the risk of overfitting. More generally, consider $bet-and-run(a_1, a_2, \ldots, a_k)$, the bet-and-run of $k$ algorithms $a_1,\ldots,a_k$. This algorithm, for a budget $b$, runs each $a_i$ with budget $b/k$:

- These $k$ runs are completely independent.
- Its random seed is split into $k$ random seeds $\omega^1, \ldots, \omega^k$.
- At the end, we choose the model with best empirical performance, i.e. $\widehat{x}$ is $\widehat{x}_i = FinalModel(\omega^i, a_i, b/k)$ minimizing $\hat{L}(\widehat{x}_i)$.

> **Theorem 4** (Bet and Run has low overfitting risk)**. *Consider* $bet-and-run(a_1, a_2, \ldots, a_k)$.*Then*
>
> $$N(\omega, bet-and-run(a_1, a_2, \ldots, a_k), b) \leq \sum_{i=1}^{k} N(\omega, a_i, b/k). \tag{A.9}$$

*Proof.* The final outcome $\widehat{x}$ is one of the $\widehat{x}_i$, so the number of possible outcomes is the sum of the numbers of possible outcomes of each algorithm individually. $\square$

Let us consider that the $a_i$ are all the same algorithm, and let us check the impact of bet-and-run on overfitting. Equation (A.9) implies a lower overfitting than with a single algorithm (corresponding to $N(\omega, a_i, b)$) if $N(\omega, a_i, b/k)$ increases more than linearly in $b/k$, which is the case for most optimization algorithms. For example, with $(1 + 1)$ evolution strategies, App. E.3 shows that $N(\omega, evolution\_strategy, b)$ is exponential in $b$.

Let us illustrate this in an example. Consider Equation (5) in the case $k_i \leq 2$, namely:

$$P_{\omega_D}\left(|\widehat{L}(\widehat{x}) - L(\widehat{x})| \geq \epsilon\right) \leq 2 \cdot \exp(\ln 2 \cdot b - 2s\epsilon^2).$$

With a bet-and-run of 3 algorithms, it becomes:

$$P_{\omega_D}\left(|\widehat{L}(\widehat{x}) - L(\widehat{x})| \geq \epsilon\right) \leq 2 \cdot \exp(\ln 2 \cdot (\log_2(3) + b/3) - 2s\epsilon^2). \tag{A.10}$$

which is better (lower) as $b \to \infty$.

### E.5 POPULATION-BASED EVOLUTION STRATEGIES (E.G. CMA-ES AND D-CMA)

For a $(\mu + 1)$ evolution strategy which stores a full ranking of the population of size $\mu$, each new point has $(1 + \mu)$ possible ranks: from best to worst in a list of cardinal $1 + \mu$, so the bound becomes $(1 + \mu)^{N-\mu}$. Extensions are possible for $(\mu, \lambda)$ or $(\mu + \lambda)$ evolution strategies:

$$N(\omega, (\mu, \lambda), a = (\mu, \lambda)\text{-ES}, b) \quad \leq \quad \binom{\lambda}{\mu}^{\frac{b-\mu}{\lambda}}$$

$$N(\omega, (\mu + \lambda), a = (\mu + \lambda)\text{-ES}, b) \quad \leq \quad \binom{\lambda + \mu}{\mu}^{\frac{b-\mu}{\lambda}}$$

In the case of CMA or D-CMA, the $(\mu, \lambda)$ strategy is more usual and used in our experiments; and typically, the population-size $\lambda$ is set as $\lambda = 4 + \lfloor 3 \log(d) \rfloor$ in dimension $d$, and $\mu$ frequently scales as $\lambda/4$ or $\lambda/2$. Sometimes there is a complete ranking of the $\mu$ selected points so that different weights are used for the $\mu$ selected points depending on their rank: this adds a factor $\mu!$ to the number of possible branches: otherwise, we get bounds as in Tab. 4.

### E.6 DIFFERENTIAL EVOLUTION (DE) AND PARTICLE SWARM OPTIMIZATION (PSO)

For differential evolution without storing the best so far, each point is just compared to its parent: we get $2^b \delta_{1,\epsilon}$: this suggests that recombination is not a problem as it does not increase the number of bits of information. When the best so far is stored, we have 3 possibilities: worse than parent, better than parent but not better than best so far, and new best so far: so the bound is $3^b \delta_{1,\epsilon}$. Similar bounds are straightforward for PSO.

### E.7 DEPENDENCE ON DATASET SIZE: SCALING AND NUMERICAL APPLICATION

App. C.2 presents classical large deviation inequalities, and the present paper shows how, in the case of comparison-based algorithms, we can derive non-vacuous bounds. Let us now see numerically how our overfitting bound behaves in practical cases.

We have focused on bounds (on the generalization loss of the model obtained by optimization) derived from a given $\delta_{1,\epsilon}$ (obtained for a single model):

We have seen in Equation (A.2) that, using Hoeffding's bound, $\delta_{1,\epsilon} = 2 \exp(-2s\epsilon^2)$ with $s = |D|$ the sample size, and $\delta \leq 2 \times (2^b) \exp(-s\epsilon^2)$, leading to $\delta$ constant when $b$ scales linearly with $s$: the acceptable number of iterations (i.e., a fixed bound on $\delta$) is linear in the sample size $D$. Applying Hoeffding, and aiming at a risk $\delta$, we get that overfitting at level $\epsilon$ is impossible if

$$b \leq \frac{\ln(\delta) + 2s\epsilon^2}{\ln(2)} - 1.$$

If we prefer the Bennett&Bernstein inequality (based on an assumption on the standard deviation, see App. C.1 (Devroye et al., 1996, Section 8.2)), we get

$$b \leq \frac{s\epsilon h_1(\epsilon/\sigma^2)}{\ln(2)} + \frac{\ln(\delta)}{\ln(2)} - 1.$$

This equation, for $s = 8000$ examples (close to GSM8k), $\delta = \frac{1}{2}$ (we are considering the median case), $\epsilon = 0.01$ (we want that precision) and $\sigma = 0.06$ (the standard variation of the loss is 0.06, we assume that the initial model is already good), leads to the claim that a budget $b \leq 91$ cannot lead to overfitting. If we assume $\epsilon = 0.04$ and $\sigma = 0.3$, we get with Hoeffding a bound 34 and with Bennett&Bernstein a bound 88. With $\epsilon = 0.06$, $\sigma = 0.3$, Bennett&Bernstein leads to a budget 189. The budget becomes 482 if we use $\epsilon = 0.1$, $\sigma = 0.3$. Also, these bounds scale linearly with $s$ (the number of user preferences in A/B testings or dataset size). These numbers are low, but we see that our bounds are not vacuous even in realistic scenarios.

## F CONSEQUENCES FOR PRIVACY AND POISONING

### F.1 PRIVACY BY DESIGN

In the context of large language models, privacy typically refers to users' desire to prevent their inputs, such as queries or conversations, from being inferred or reconstructed. Several studies have shown that gradient-based training methods can leak such information (Zhu et al., 2019; Hitaj et al., 2017; Haim et al., 2022; Carlini et al., 2021).

In contrast, the *BBoxER* approach is an instance of **privacy by design** (Gershoff, 2025): It does not rely on gradients, nor does it require access to the raw content of user sessions. Instead, it only leverages a binary satisfaction signal–whether a user prefers one version of the model over another. This can be collected via simple A/B testing, or from the individual samples of the dataset by comparing the models over the singleton {sample}. Then the algorithm proceeds using aggregated comparison outcomes. In particular, for A/B testing, our method operates without knowledge of the specific queries or model responses, that do not need to leave the user's device, and in the general case, we only feed the algorithm the aggregate of model comparison without direct access to the underlying data.

*BBoxER* privacy guarantees can be framed in terms of differential privacy (Dwork, 2008), which ensures that the output of a training algorithm does not change significantly when a single data point

is modified. Given two datasets $D_1$ and $D_2$ differing by a single element, differential privacy requires that for any output set $S$,

$$P(A(D_1) \in S) \le \exp(\varepsilon) \cdot P(A(D_2) \in S) + \delta. \tag{A.11}$$

We now examine how this applies in two distinct cases: the privacy of user queries and responses and the general case of privacy.

**Privacy of Queries and Responses.** (like in A/B testing setting) Our algorithm never observes the actual content of user interactions. If $D_1$ and $D_2$ differ by one prompt or answer or any data unrelated to preference signals, then the outcome of Alg. 1 remains unchanged. In that case, the algorithm behaves identically on both datasets, satisfying differential privacy with $\varepsilon = 0$, $\delta = 0$. This guarantee is stronger than what is typically offered in federated learning: since no gradients are shared or aggregated, the system is inherently robust to inference attacks such as those in (Zhu et al., 2019; Hitaj et al., 2017; Carlini et al., 2021).

**Algorithm stability.** (general case of applying BBoxER on a dataset $D$) We may also be interested in the influence of a single satisfaction response $r_i$ associated with a specific user, or in the case were we compute this preference signal from the data. In this setting, we consider the case where $D_1$ and $D_2$ differ by exactly one such response, altered during a run. Applying Equation (7) with $m = 1$, we get that $A(D_1)$ and $A(D_2)$ are the same with probability at least $1 - 3b/\sqrt{2s\pi}$ when we have $s$ data points and $b$ iterations. Which represents $\varepsilon = 0$ and $\delta = 3b/\sqrt{2s\pi}$ in Equation (A.11) but is insufficient from a DP standpoint given that $\delta$ needs to be polynomial in $\frac{1}{s}$. A differentially private version of BBoxER can be obtained by adding a Gaussian noise to every user preference signal in case of A/B testing, or by adding a Gaussian noise to the decision metric (like accuracy) in case we compute the performance from the prompts/outputs of the model.

**Comparison with Other Approaches.** In supervised fine-tuning, both inputs and outputs are required for training. Reinforcement learning additionally requires access to the model's responses and reward signals. While federated learning aims to limit privacy leakage by aggregating gradients locally, it still exposes sensitive intermediate data—such as gradients—that have been shown to be vulnerable to attacks (Hitaj et al., 2017). In contrast, our approach bypasses these vulnerabilities entirely by never interacting with such information in the first place, in the case of A/B testing. Or, by limiting the impact of individual samples through an aggregate compressed signal from the data.

## F.2 POISONING

Poisoning attacks seek to influence the outcome of a learning algorithm by injecting malicious data. This threat has been widely studied in machine learning (Biggio et al., 2012; Pathmanathan et al., 2025; Wan et al., 2023), and it is particularly relevant in interactive settings such as retrofitting. In our context, a poisoning attempt manipulates up to $m$ user preferences per iteration to bias the optimization process.

Our robustness Theorem 2 quantifies the impact of such attacks: the probability that the final model is altered remains bounded by $\frac{(2m+1)b}{\sqrt{2s\pi}}$. This implies that an adversary must control $m \sim \sqrt{s}/b$ user preferences per round to affect the result significantly. Such an attack becomes increasingly difficult as the number of users (or dataset size) $s$ grows.

Importantly, this guarantee holds even under a strong adversarial model (Kearns & Li, 1993), where the attacker is computationally unbounded and has full knowledge of the system. In practice, this highlights the resilience of our retrofitting method to small-scale poisoning.

# G ADDITIONAL THEORETICAL RESULTS AND PROOFS

## G.1 PROOF OF THEOREM 2

We first prove the following Lemma.

**Lemma 1** (Privacy and robustness to poisoning – single comparison). *Consider a frequency $f = \frac{1}{s}\sum_{i=1}^{s} r_i$ computed by considering $s$ samples $r_1, \ldots, r_s$ independent and identically distributed*

with $P(r_i = 1) = f_0$ and $P(r_i = 0) = 1 - f_0$ for some unknown $f_0$. Then

$$P_{r_1,\ldots,r_s}\left(\exists\, f' \text{ such that } |f' - f| \leq m/s \text{ and } (f - \frac{1}{2})\cdot(f' - \frac{1}{2}) \leq 0\right) \leq \frac{2m + 1}{\sqrt{2s\pi}}. \quad (A.12)$$

*Proof.* First, using (Devroye et al., 1996, Lemma A.3), we observe that

$$\forall x \in \{0, \ldots, s\}, P(B(s, \frac{1}{2}) = x) \leq \frac{1}{\sqrt{2s\pi}}. \quad (A.13)$$

Consider a probability $p$ that individual comparisons are in favor of a model $model_1$ against a model $model_2$. Assume that a modification of the training set has an impact $\epsilon$ on the frequency $f$ with which $model_1$ was preferred against model $model_2$: the new frequency $f'$ is in $[f - \epsilon, f + \epsilon]$. $(f - 1/2) \times (f' - 1/2) \leq 0$ is possible only if $f \in [1/2 - \epsilon, 1/2 + \epsilon]$. And since $f \sim \frac{1}{s}B(s, p)$, $P(f \in [1/2 - \epsilon, 1/2 + \epsilon])$ is maximum if $p = 1/2$. Then,

$$P\left(f \in [1/2 - \epsilon, 1/2 + \epsilon]\right) \leq P\left(\frac{1}{s}B(s, 1/2) \in [1/2 - \epsilon, 1/2 + \epsilon]\right)$$

$$\leq P\left(B(s, 1/2) \in [s(1/2 - \epsilon), s(1/2 + \epsilon)]\right)$$

$$\leq \delta := \frac{2s\epsilon + 1}{\sqrt{2s\pi}} \text{ thanks to Equation (A.13)}.$$

If someone modifies the data by modifying the answers corresponding to $m$ users, the frequency moves to $f'$ instead of $f$ with $|f - f'| \leq \epsilon = m/s$ where $s$ is the number of independent users/samples. This shows Equation (A.12). □

To show Theorem 2, we apply the Bonferroni correction when $m$ data values are modified at each iteration and we have $b$ iterations. Then, the probability of a difference between $output_1$ and $output_2$ is upper bounded by $b \times \delta$. This shows Equation (7).

### G.2 THEOREM ON DATA EXTRACTION AND PROOF OF COR. 2

Cor. 2 is a direct consequence of the following Theorem.

---

**Theorem 5.** *The cardinality of the set of possible outputs of* BBoxER *for a given* $m_0, \omega, a, b$ *is bounded as follows:*

$$\#\{\text{BBoxER}(m_0, \omega, D_i, a, b); i \in \{1, \ldots, n\}\} \leq \sup_D \prod_{i=1}^{b} k_i. \quad (A.14)$$

---

*Proof.* Equation (A.14) is an immediate consequence of Equation (1). If $A(m_0, D) := BBoxER(m_0, \omega, D, a, b)$, then vulnerability to data extraction as in Equation (8) immediately implies that $\{E(A(m_0, D_i)); i \in \{1, \ldots, n\}\}$ has cardinal at least $n$, and therefore $\{A(m_0, D_i); i \in \{1, \ldots, n\}\}$ has cardinal at least $n$. If $2^b > n$ and $k_i \leq 2$, this contradicts Equation (A.14). □

This means that no matter how large $s$ is, retrofitting can only produce a number of different outputs upper-bounded by the product over the $k_i$, which represent the number of possible data-dependent choices per iteration - e.g., $2^b$ for algorithms with $k_i \leq 2$ and $b$ iterations.

## H EXPERIMENTAL DETAILS

Here we provide details for the settings of the experiments in Sec. 5. The licenses for all datasets used in this work can be found in Tab. 5. Experiments were implemented using Lingua (Videau et al., 2024a) and vLLM (Kwon et al., 2023).

All BBO algorithms are used with their default parameters in Nevergrad (Rapin & Teytaud, 2018). At each iteration of Alg. 1, the modified model is evaluated on GSM8k train (8-shot, unless otherwise

Table 5: Datasets licenses

| License | Datasets |
|---|---|
| MIT License | GSM8k, MATH, MATH500, Hellaswag, SVAMP |
| CC BY-SA 4.0 | GSM+, ARC Easy, ARC Challenge |

specified) and the objective function to optimize is the exact match. At the end of each run, the *FinalModel* is evaluated on the same objective function. Specifically, we compute the performance of $modified(m_0, \hat{x})$ where $\hat{x}$ is the update that reaches the best objective function value during training.

**Experiment 1.** We apply *BBoxER* (Alg. 1) with rank-1 (LoRA) multiplicative parameter updates on the *unembedding layer*, as detailed in App. B. In both base and instruct models, the total number of parameters to optimize is $hidden\_dim + vocab\_size = 4096 + 128, 256 = 132, 352$. Each iteration requires around 40s on 16 A100 GPUs.

**Experiment 2.** We explore 3 setups on Llama3.1-8B:

1. **First attention layer**, where we perform a broadcast multiplicative update to the Q matrix in the first attention layer by broadcasting a vector of size 4096 to the $4096 \times 4096$ Q matrix and then using the update formula as in App. B.1.

2. **All attention layers**, where we perform a broadcast multiplicative update on all the $Q$ matrices of all attention layers. Resulting in $n\_layers \times hidden\_dim = 32 \times 4096 = 131, 072$ parameters to optimize, using the update formula as in App. B.2

3. **All attention layers and no few-shot**, where we apply a broadcast multiplicative update on all the $Q$ matrices of all attention layers, as in 2 above. However, we do not include any few-shot examples in the prompt during either training or testing.

All experiments are carried out with budget $b = 150$ unless otherwise specified. In addition to GSM8k test, we also evaluate the results on GSM+ (8-shot) and SVAMP (8-shot). Each iteration requires around 40s on 16 A100 GPUs.

**Experiment 3** In this experiment, D-CMA is used to directly optimize all normalization layers of Llama3.1-8B(-Instruct) and Qwen2.5-3B-Instruct (without using any low rank modification). The number of updated parameters is $n\_norm\_layers \times hidden\_dim = 65 \times 4096 = 266, 240$ for Llama3.1-8B(-Instruct) and $n\_norm\_layers \times hidden\_dim = 73 \times 2048 = 149, 504$ for Qwen2.5-3B-Instruct. All experiments are conducted with budget $b \in \{150, 300\}$. Furthermore, we evaluate on additional mathematical reasoning benchmarks: MATH500 (4-shot), Hellaswag (0-shot), GSM+ test_mini (8-shot), ARC Easy (0-shot) and ARC Challenge (0-shot), AMC23 (0-shot). Each iteration requires 30s on 16 A100 GPUs for Llama3.1-8B(-Instruct) and 25s on 16 A100 GPUs for Qwen2.5-3B-Instruct.

**Experiment 4** Most MIA rely on the loss (negative likelihood) of the model over the tested sample to determine membership in the training dataset (Fu et al., 2023; Carlini et al., 2022; 2021). We can formulate the membership test as follows:

$$f\left(\text{NLL}\{sample\}\right) > \text{threshold}\{sample\}$$

where $f$ denotes a transformation (usually continuous) applied to the sample loss. The threshold used in membership test is sample dependent, in order to account for the fact that some samples may exhibit low losses even without being part of the training set, simply because they share patterns that were seen during training.

For example, in Carlini et al. (2022), the authors propose to model the distribution of model's "confidence" with a gaussian distribution; they train shadow models to estimate the parameters $(\mu_{in}, \sigma_{in}, \mu_{out}, \sigma_{out})$ and perform a Likelihood-ratio test, with $p$ the density function of Gaussian

distribution and $\phi$ the logit scaling as defined in the LiRA paper $\phi(t) = \log(\frac{t}{1-t})$:

$$\frac{p(\phi(\text{NLL}\{sample\}); \mu_{in}, \sigma_{in})}{p(\phi(\text{NLL}\{sample\}); \mu_{out}, \sigma_{out})} > \text{threshold}$$

$$\Leftrightarrow p(\phi(\text{NLL}\{sample\}); \mu_{in}, \sigma_{in}) > \text{threshold} \times p(\phi(\text{NLL}\{sample\}); \mu_{out}, \sigma_{out})$$

Therefore, if a learning method does not impact the loss of the model over each sample, it would be difficult or impossible to perform any MIA that relies on statistical tests based on the loss. We thus propose the two following metrics as proxies for the ability of membership inference attacks to succeed:

$$\frac{1}{\text{num of samples}} \sum_{sample} |\text{per\_token\_NLL}_{\text{new model}}\{sample\} - \text{per\_token\_NLL}_{\text{initial model}}\{sample\}| \quad \text{(A.15)}$$

$$\frac{1}{\text{num of samples}} \sum_{sample} |\text{per\_token\_NLL}^{CoT,a}_{\text{new model}}\{sample\} - \text{per\_token\_NLL}^{CoT,a}_{\text{initial model}}\{sample\}| \quad \text{(A.16)}$$

The only distinction between Equation (A.15) and Equation (A.16) is which tokens are used to compute the negative log-likelihood (NLL): Equation (A.16) computes NLL only over the (chain-of-thought, answer) pair, whereas Equation (A.15) computes NLL over the entire input (few-shot examples, question, chain-of-thought, and answer). We aggregate the absolute differences (i.e., sum of absolute changes in sample NLL) across all samples so that positive and negative changes do not cancel each other out; although in practice the loss of the new finetuned model will be lower than the initial model. To interpret the metric: a low value indicates that the sample losses are only slightly affected by the learning method, suggesting it would be difficult to mount an attack that relies on loss changes, whereas a large value indicates that it might be possible (but not certain) to perform such an attack successfully. By showing in Table Tab. 2 that *BBoxER* barely affects these metrics, and is orders of magnitude less than finetuning methods, including privacy-aware ones like DP-AdamW, we demonstrate robustness of *BBoxER* against a vast majority of membership inference attacks.

Below we detail all algorithms used in Tab. 2:

- **ft**: default SFT with next token prediction loss, global batch size of 128, same template used during evaluation that includes few shots examples but only the CoT and the answer contribute to the loss. AdamW is used with ($beta_1 = 0.9, beta_2 = 0.95$) and a learning rate of $1e^{-5}$ with cosine annealing and warm-up over the first 5% steps. The training is done for 2 and 5 epochs.

- **ft-norm**: similar to **ft**, but only the parameters of RMSNorm layers are optimized, and the learning rate is increased to $2e^{-4}$ instead.

- **DP-AdamW**: differentially private training with gradient norm clipping set to $1.0$, $\delta = 10^{-5}$, $L = 128$, $N = 7473$ samples, 292 gradient steps (5 epochs) and a target $\epsilon = 8$, the privacy accounting of (Abadi et al., 2016) is used to infer the added noise scale $\sigma$. The training is done with the same template used during evaluation that includes few shots, all tokens contribute to the loss.

- ***BBoxER*-norm**: exact setup as in experiment 3 for our algorithm, step $\delta = 0.01$, only updating RMSNorm layers with an exponential multiplicative full update. Different budget values are used 150, 300, 1200. Nevergrad seed = 42.

# I ADDITIONAL EXPERIMENTS

In this section, we describe extensions of the experiments performed in Sec. 5, in particular for Bet-and-Run variants and for training on larger datasets.

**Extensions of Experiment 2.** We have performed additional experiments on Llama3.1-8B in the spirit of Experiment 2 (Sec. 5), by modifying the *output layer* instead of the attention layers, using a multiplicative update by broadcasting a vector of size $vocab\_size = 128,256$ as described in App. B.1. We also experiment with two different update constants in $\{0.01, 0.0001\}$. Tab. 6 shows the results. We see improvements particularly for the larger update constant and Bet-and-Run algorithms.

Table 6: **Additional Experiments**. Counterpart of Tab. 1 for the output layer: Performance difference (%) after vs before *BBoxER* with Llama3.1-8B (Base) on GSM8k-train, evaluated on GSM8k-test (ID), and (OOD) SVAMP and GSM+, with budget $b = 150$. Number of seeds in brackets. Red: detrimental runs, $< -1\%$. Green: beneficial runs, $> 1\%$. Blue rows are bet-and-run cases.

| BBO algorithm (# runs) | GSM progress | SVAMP progress | GSM+ progress |
|---|---|---|---|
| Output layer, no few-shot, broadcast d=128256 $w0 \times e(0.0001 \times x)$ | | | |
| Base score | 20.31 | 58.30 | 19.06 |
| OneFifth (2) | $0.49 \pm 0.34$ | $0.05 \pm 0.15$ | $0.01 \pm 0.02$ |
| D-CMA | 0.07 | 0.00 | 0.01 |
| Lengler | 0.00 | 0.00 | 0.00 |
| COLengler (2) | $0.00 \pm 0.00$ | $0.05 \pm 0.05$ | $0.00 \pm 0.00$ |
| Output layer, broadcast d=128256 $w0 \times e(0.01 \times x)$ | | | |
| Base score | 56.93 | 76.6 | 51.34 |
| Lengler (2) | $1.89 \pm 1.44$ | $-0.2 \pm 0.6$ | $0.9 \pm 0.41$ |
| COLengler (5) | $2.01 \pm 0.49$ | $-0.76 \pm 1.03$ | $0.19 \pm 0.43$ |
| Discrete (4) | $0.018 \pm 0.03$ | $0.075 \pm 0.08$ | $0.005 \pm 0.015$ |
| Uniform-Disc (2) | $0.79 \pm 0.11$ | $0.45 \pm 0.15$ | $-0.05 \pm 0.05$ |
| 3-OneFifth | 2.27 | 0.00 | 0.57 |
| 3-Disc | 1.89 | 0.20 | -0.27 |
| 3-Disc | 2.19 | -2.20 | -0.02 |

**Experiment 3 for Bet-and-Run Algorithms.** The theoretical results in App. E.4 demonstrate that Bet-and-Run algorithms enable the use of larger budgets without risks of overfitting. This has been partially validated in Experiment 2 by the results of 3-OneFifth, performing well even with greater budget (Table 1). We here extend this analysis and report Bet-and-Run results for the setup of Experiment 3 in Tab. 7. Specifically, we run *BBoxER* under the same setup as in Experiment 3 (detailed in App. H), and use the BetAndRun (N, b) variant of D-CMA, where we run D-CMA for N different runs with budget $b' = N/b$, and pick the update $\hat{x}$ that performed the best across these runs. In practice, we use the runs with 5 different seeds from Tab. 3 and report the mean and standard deviation of all possible combinations $\binom{5}{N}$ with $N \in \{3, 4\}$.

Table 7: **Extension of Experiment 3 to Bet-and-Run algorithms**, same setting as Tab. 3

| Models | | ID | OOD | | | | | |
|---|---|---|---|---|---|---|---|---|
| | | GSM8k | GSM+ | MATH500 | ARC Easy | ARC Challenge | AMC23 | Hellaswag |
| Llama3.1-8B | | 54.97 | 36.50 | 20.20 | 83.04 | 55.19 | 0.00 | 80.71 |
| Llama3.1-8B-BBoxER-BetAndRun | (N=3, b=450) | 55.66±0.42 | 37.62±0.34 | 20.04±0.45 | 82.96±0.08 | 55.03±0.03 | 3.00±3.67 | 80.62±0.05 |
| | (N=3, b=900) | 56.83±0.21 | 37.74±0.28 | 20.36±0.77 | 83.26±0.11 | 54.96±0.18 | 1.00±1.22 | 80.68±0.11 |
| | (N=4, b=600) | 55.62±0.21 | 37.73±0.30 | 20.20±0.40 | 82.94±0.03 | 55.02±0.00 | 1.50±3.00 | 80.60±0.04 |
| | (N=4, b=1200) | 56.85±0.12 | 37.77±0.22 | 20.48±0.64 | 83.29±0.10 | 54.88±0.07 | 0.50±1.00 | 80.71±0.10 |
| Llama3.1-8B-Instruct | | 77.26 | 54.37 | 37.60 | 79.62 | 55.45 | 22.50 | 79.90 |
| Llama3.1-8B-Instruct-BBoxER-BetAndRun | (N=3, b=450) | 78.20±0.21 | 55.02±0.12 | 36.68±0.36 | 78.90±0.38 | 55.40±0.08 | 23.25±2.25 | 79.96±0.04 |
| | (N=3, b=900) | 78.37±0.21 | 55.18±0.15 | 37.20±0.54 | 79.57±0.12 | 55.59±0.19 | 18.00±1.50 | 79.92±0.02 |
| | (N=4, b=600) | 78.26±0.03 | 55.03±0.10 | 36.52±0.24 | 78.77±0.34 | 55.42±0.07 | 24.00±2.00 | 79.96±0.03 |
| | (N=4, b=1200) | 78.45±0.18 | 55.17±0.08 | 37.04±0.48 | 79.54±0.10 | 55.54±0.17 | 17.50±0.00 | 79.91±0.02 |
| Qwen-2.5-3B-instruct | | 79.98 | 62.29 | 41.00 | 72.39 | 47.38 | 32.50 | 74.94 |
| Qwen-2.5-3B-instruct | (N=3, b=450) | 83.06±0.24 | 62.12±0.32 | 42.70±0.20 | 72.69±0.12 | 47.85±0.17 | 36.00±1.22 | 74.99±0.07 |
| | (N=3, b=900) | 83.90±0.42 | 61.66±0.32 | 42.26±0.65 | 72.61±0.21 | 47.63±0.31 | 37.00±2.45 | 75.04±0.05 |
| | (N=4, b=600) | 83.08±0.18 | 62.24±0.27 | 42.72±0.16 | 72.71±0.05 | 47.86±0.10 | 35.50±1.00 | 75.00±0.06 |
| | (N=4, b=1200) | 84.08±0.30 | 61.55±0.02 | 42.52±0.16 | 72.68±0.19 | 47.52±0.27 | 36.00±2.00 | 75.06±0.04 |

Comparing the results in Tab. 7 with those in Tab. 3 (*BBoxER* without Bet-and-Run), we observe that in 100% runs (with or without Bet-and-Run) *BBoxER* improves results; in 83% of cases Bet-and-Run outperforms its base counterpart.

**Experiment 3 for Larger Training Sets.** In Tab. 8 we report the results of running *BBoxER* with a larger training dataset. This experiment is motivated by the theoretical prediction that the allowable budget scales linearly with the dataset size. Formally, we adopt the same experimental setup as in Experiment 3 (see App. H), but in addition to GSM8k train (7,473 samples), we also include the MATH train set (7500 samples) into the training set, thus doubling its size compared to previous experiments. We define the objective function for *BBoxER* to be the average of exact matches (EM) across both datasets : $metric = \frac{1}{2}EM(GSM8k) + \frac{1}{2}EM(MATH)$. We run *BBoxER* with a budget of up to $b = 1200$ and evaluate the model every 150 iterations. As the budget increases, we observe

eventual overfitting. Interestingly, we find that the Llama3.1-8B base model does not overfit on the GSM8k test set and continues to improve, contrary to its instruct variant, which begins to overfit earlier. See also Figure 3 for illustration of performance gain on some of the benchmarks.

Table 8: **Experiment 3 (full update, normalization layers) on larger training set.** Comparison of model performance of BBoxER run with D-CMA across various benchmarks, 1 seed.

| Models | | ID | | | OOD | | | | |
|---|---|---|---|---|---|---|---|---|---|
| | | GSM8k | MATH | MATH500 | GSM+ | ARC Easy | ARC Challenge | AMC23 | Hellaswag |
| Llama3.1-8B | | 54.97 | 21.22 | 20.20 | 36.50 | 83.00 | 55.19 | 0.00 | 80.71 |
| | (b=150) | 55.42 | 21.00 | 19.20 | 36.96 | 83.09 | 54.85 | 0.00 | 80.76 |
| | (b=300) | 55.27 | 21.14 | 18.80 | 36.79 | 83.09 | 54.85 | 2.50 | 80.78 |
| | (b=450) | 55.27 | 21.14 | 18.80 | 36.79 | 83.09 | 54.85 | 2.50 | 80.78 |
| | (b=600) | 55.42 | 20.74 | 20.60 | 37.67 | 83.04 | 54.94 | 5.00 | 80.77 |
| Llama3.1-8B-BBoxER | (b=750) | 57.85 | 20.92 | 20.20 | 37.67 | 83.00 | 54.85 | 0.00 | 80.83 |
| | (b=900) | 57.47 | 21.58 | 21.20 | 38.75 | 82.66 | 55.11 | 0.00 | 80.66 |
| | (b=1050) | 57.47 | 21.58 | 21.20 | 38.75 | 82.66 | 55.11 | 0.00 | 80.66 |
| | (b=1200) | 58.61 | 21.14 | 21.00 | 39.33 | 83.04 | 55.36 | 5.00 | 80.81 |
| Llama3.1-8B-Instruct | | 77.26 | 38.84 | 37.60 | 54.37 | 79.62 | 55.36 | 22.50 | 79.90 |
| | (b=150) | 77.41 | 38.78 | 37.40 | 54.12 | 79.45 | 55.36 | 25.00 | 80.03 |
| | (b=300) | 78.24 | 38.86 | 38.60 | 55.04 | 79.58 | 55.36 | 30.00 | 79.91 |
| | (b=450) | 77.56 | 38.40 | 37.00 | 54.62 | 80.00 | 55.54 | 22.50 | 79.94 |
| | (b=600) | 77.79 | 39.04 | 38.20 | 54.58 | 79.83 | 55.54 | 22.50 | 79.94 |
| Llama3.1-8B-Instruct-BBoxER | (b=750) | 78.70 | 38.86 | 37.00 | 55.33 | 79.45 | 55.36 | 15.00 | 80.00 |
| | (b=900) | 78.54 | 38.30 | 37.40 | 55.25 | 79.70 | 55.45 | 20.00 | 80.10 |
| | (b=1050) | 77.86 | 38.62 | 37.40 | 55.29 | 79.53 | 55.54 | 20.00 | 79.98 |
| | (b=1200) | 77.79 | 38.86 | 37.40 | 55.42 | 79.87 | 55.79 | 15.00 | 80.00 |
| Qwen2.5-3B | | 79.98 | 43.68 | 41.00 | 62.38 | 72.35 | 47.38 | 32.50 | 74.94 |
| | (b=150) | 83.55 | 43.32 | 39.80 | 62.00 | 72.77 | 48.24 | 37.50 | 75.07 |
| | (b=300) | 82.79 | 44.10 | 42.60 | 62.17 | 72.60 | 47.64 | 40.00 | 75.09 |
| | (b=450) | 83.09 | 43.66 | 39.40 | 62.58 | 72.81 | 48.24 | 42.50 | 74.94 |
| | (b=600) | 82.64 | 44.94 | 42.60 | 62.17 | 72.98 | 47.90 | 42.50 | 74.93 |
| Qwen2.5-3B-BBoxER | (b=750) | 82.87 | 45.26 | 44.00 | 61.54 | 71.54 | 47.47 | 32.50 | 74.85 |
| | (b=900) | 84.08 | 45.68 | 47.80 | 63.04 | 72.60 | 48.15 | 40.00 | 74.92 |
| | (b=1050) | 83.17 | 46.50 | 45.60 | 62.58 | 72.85 | 47.47 | 32.50 | 74.98 |
| | (b=1200) | 82.49 | 46.72 | 45.80 | 61.46 | 73.32 | 46.27 | 37.50 | 74.91 |

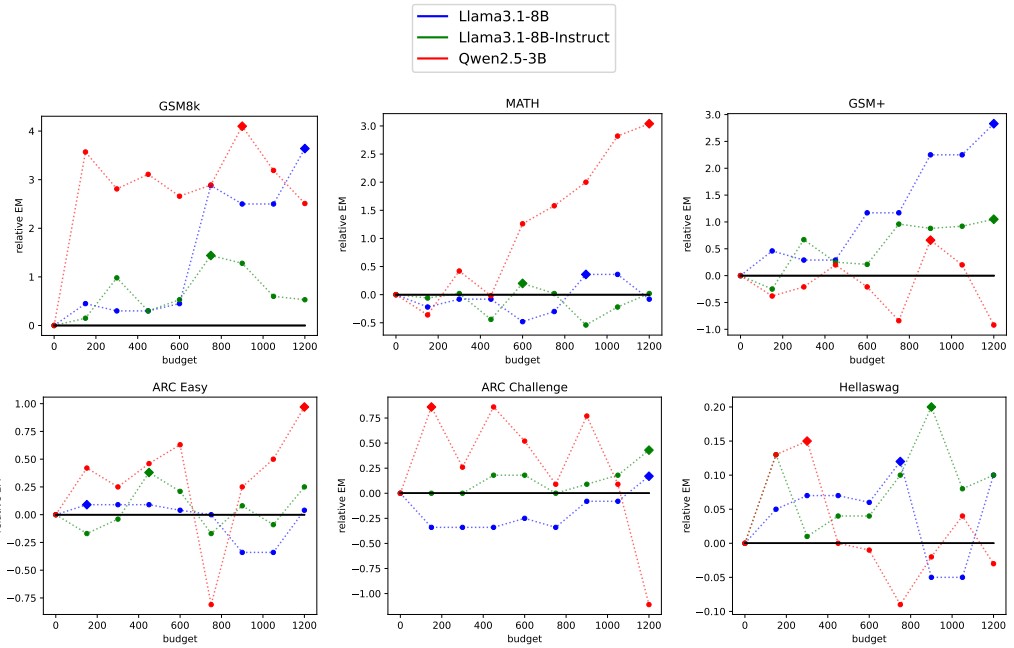

Figure 3: **(full update, normalization layers)** Comparison of model performance of BBoxER run with D-CMA across various benchmarks, 1 seed. See Tab. 8.

Table 9: **Experiment 3 (full update, normalization layers) different tasks.** Relative accuracy improvement using *BBoxER* run with D-CMA, 5 seeds.

| *BBoxER* budget b | ARC Easy | ARC Challenge | Hellaswag |
|---|---|---|---|
| 150 | $0.01 \pm 0.10$ | $-0.14 \pm 0.12$ | $0.16 \pm 0.07$ |
| 300 | $0.13 \pm 0.06$ | $-0.21 \pm 0.17$ | $0.25 \pm 0.11$ |
| 600 | $0.18 \pm 0.16$ | $-0.05 \pm 0.25$ | $0.47 \pm 0.06$ |
| 1200 | $0.85 \pm 0.16$ | $0.24 \pm 0.21$ | $0.89 \pm 0.13$ |

**Experiment 3 for different tasks.** In Tab. 9 we report the results of running *BBoxER* on tasks beyond mathematical reasoning. We update Llama-3.1-8B on the ARC-Easy training split and evaluate on the ARC-Easy test split, and repeat the same procedure for ARC-Challenge and Hellaswag. We report the mean and standard deviation across five random seeds of the relative improvement, or difference of accuracy, compared to the base model. We notice that on ARC Easy and Hellaswag we get an improvement, while it is not the case on ARC Challenge. This likely stems from the base model's performance across tasks. On ARC Easy and Hellaswag, the base model scores about 80%, so "retrofitting" can provide meaningful improvement. In contrast, on ARC-Challenge the base is about 50% (random is at 25%), and BBoxER is designed to provide additional improvement as a lightweight add-on, rather than heavy duty training: if the base is weak, perturbing parameters won't meaningfully improve the model even if we notice slight improvement on the training split.

**Further Discussion of all Experiments.** Consistent with our theoretical results, our results are robust and show performance improvement with little to no overfitting even with larger budgets. We note that even when running various experiments with randomly chosen parameters, significant performance drops are rare: In Figure 1 and in Tab. 1, all data points are independent from one another (i.e. computed independently); This allows us to compute P-values (App. J) to support our claims of statistical significance of the observed performance improvements.

In order to contextualize our findings, we can also compare to post-training with GRPO which proved to be highly effective on reasoning tasks. Most works deploy GRPO on larger training sets and we were unable to find results for the models deployed in our study. However, as shown in (Lin et al., 2025), GRPO achieved a notable 22% improvement when training a Qwen-2.5-1.5B-Instruct model on the GSM8k dataset, leading to results between 77% and 81% with 1B models, without transfer to OOD. However, such improvement jumps tend to shrink with larger models. Nevertheless, though the Qwen-2.5-3B-Instruct model we use already starts at nearly 80%, it is improved to 84% by *BBoxER* (Tab. 7).

There are only few works that train on GSM8k and study OOD transfer. The authors of (Sun et al., 2025) employ an 'on-policy' training approach, generating training samples through model prompting and subsequently fine-tuning on the resulting question-answer pairs. Their approach yields significant gains on both GSM8k and GSM+ benchmarks when evaluated in a *zero-shot* setting, by up to 28.8% on GSM8k with Llama3.1-8B-Instruct. However, the gains are less pronounced in the 5-shot setting, with an increase of 1.8% using self-consistency and 5 generations, figures that are close to those obtained by *BBoxER* on GSM8k and GSM+, as can be seen in Tab. 7.

In any case, remember that our goal in this work is not to achieve state-of-the-art performances on these and other benchmarks, but rather to show that our approach sacrifices little for provable privacy and overfitting guarantees.

## J  P-VALUES

Contemporary papers on LLMs frequently involve a small number of experiments, due to the significant computational burden. But because we run independent tests, we can compute rigorous p-values by Fisher's exact test, getting a measure of statistical significance despite the moderate number of experiments:

- We computed the *p*-value of the frequency of green vs red in the columns of Tab. 1 and Tab. 6, the null hypothesis being "the probability of improvement is $\leq 0.5$".

- For the values in Figure 1, we computed the $p$-value for the obtained frequency of successful runs (i.e., runs above baseline, vs below baseline) for budget $> 20$, under the null hypothesis "the probability of improvement is $\leq 0.5$".

- In Experiment 3, we run *BBoxER* with budget $b = 150$ or $b = 300$. For each column of results we performed an exact Fisher test on all *BBoxER* runs (both 150 and 300, without considering the Bet-And-Runs) and computed p-values: there is therefore one $p$-value per downstream task, against the null hypothesis "there is a probability of improvement $\leq \frac{1}{2}$". These different tests are not independent.

When computing p-values from tables of results, we used the initial data, not the averaged ones, obtaining one data point per independent run.

**P-values Values.**

**Experiment 1**: P-values in Figure 1 are $3e^{-8}$ (left), $3e^{-5}$ (right).

**Experiment 2**:

- Tab. 1: P-values are significant for GSM ($5e^{-3}$) and GSM+ (0.03), not for SVAMP.

- Tab. 6: P-values are 0.003 for GSM, 0.5 for the transfer to GSM+, no improvement for SVAMP.

**Experiment 3**: The P-values are $5e^{-7}$ for GSM8k, $1e^{-4}$ for GSM+, $1e^{-4}$ for AMC23.
Other results are usually positive but without significant p-value: 0.13 for ARC Easy, 0.21 for ARC Challenge, 0.36 for MATH500, 0.12 for Hellaswag.

## K   BBoxER EXECUTION EXAMPLE

In this section we provide a step by step execution example of *BBoxER* (Alg. 1), using OneFifth (Alg. 2, (1+1)-ES) as the BBO algorithm and *Full Layer Update* on the normalization layers as our choice of $modified$.

**Input**: seed $\omega$, dataset $D$, BBO algorithm $a$=OneFifth, budget $b$

**step 1**: initialize the internal state by seeding the rng with $\omega$, and fixing the step size $\sigma$

**step 2**: sample a $z \sim \sigma \mathcal{N}(0, I)$ via $a.ask()$

**step 3**: update model weights $m_i = modified(m_0, z)$, where every RMSNorm layer weights $w_i^{(j)}$ are updated with $w_i^{(j)} = w_0^{(j)} \times \exp(0.01 * z_j)$ and $z = (z_1, \ldots, z_{\text{number RMSNorm layers}})$

**step 4**: $k_i = a.numCases() = 2$ (we only compare with the best so far, therefore 2 possible choices)

**step 5**: $choice_i = a.compare((m_{best}, m_i), D)$ this comparison is either done by majority vote in A/B testing or by comparing the performance of $m_{best}$ and $m_i$ over $D$ (note that only the comparison result is needed and not the actual performance of each model)

**step 6**: $a.tell(choice_i)$ and $I_i = (I_{i-1}, choice_i)$ update the internal state of algorithm a : if $m_i$ is better ($choice_i = 1$) then $\sigma = 2 * \sigma$, otherwise ($choice_i = 0$) $\sigma = 2^{-\frac{1}{4}} * \sigma$

-> repeat steps 2 to 6 for b iterations

## L   LLM USAGE

We acknowledge the use of large language models (LLMs) to assist with text checking for clarity and grammar, as well as for figure editing.

