# OpenReview forum: "Tuning without Peeking: Provable Privacy and Generalization Bounds for LLM Post-Training"
_ICLR.cc/2026/Conference — Submitted to ICLR 2026_

### Official Review · Reviewer_PoZh · 2025-10-27

**Soundness:** 4
**Presentation:** 3
**Contribution:** 3
**Rating:** 8
**Confidence:** 3

**Summary:**

This paper introduces BBoxER, a novel black-box post-training framework that enables secure fine-tuning of large language models without direct access to gradients or training data. Instead of learning from full supervision, BBoxER compresses the training signal into discrete comparison traces, transmitting only a few bits of feedback per iteration. Leveraging black-box optimizers such as CMA-ES, OneFifth, and D-CMA, it updates model parameters solely based on relative performance comparisons. Theoretically, the method establishes provable generalization bounds (via Hoeffding and Bonferroni inequalities) and robustness guarantees under data poisoning and differential privacy constraints (Theorem 2). Experiments on GSM8K, SVAMP, and GSM+ show competitive reasoning improvements despite the extremely limited information flow. Conceptually, BBoxER reframes model fine-tuning as information-constrained optimization, providing a principled bridge between privacy, robustness, and efficient adaptation in large-scale LLMs.

**Strengths:**

1. A novel research perspective of black-box retrofitting enables fine-tuning large language models without accessing gradients or original training data.
2. The method tightly integrates the information compression bottleneck with generalization theory, formally bounding the information flow to one bit per iteration.
3. Experiments on mathematical reasoning tasks (GSM8K, SVAMP, GSM+) show 5–7% accuracy gains under a no-gradient setting, confirming the method’s effectiveness.

**Weaknesses:**

1. Theoretical derivations contain omissions and jumps; Theorem 2 lacks a complete proof and key assumptions are not clearly explained.
2. The correspondence between theory and implementation is unclear; the link between continuous Gaussian perturbations and discrete branching assumptions should be clarified.
3. In Appdex C.2.1, the symbol $\pi_i$ is not explicitly defined in the paper, causing confusion and hindering understanding..
4. The code is not released and implementation details are missing, making the method hard to follow and reproduce.
5. The experimental scope is narrow, focusing only on mathematical reasoning tasks without broader task validation.
6. The privacy analysis is mostly qualitative; although extraction and membership inference attacks are discussed, there is no empirical support or comparison.
7. Several references are duplicated and should be cleaned up for clarity and consistency.

**Questions:**

See weakness.
1. Can simple Gaussian perturbations of model parameters be effectively captured by the theoretical generalization and robustness bounds? If so, how do they correspond to the finite branching assumption in the theory?
2. Why does the paper adopt an exponential form of parameter update (e.g., $exp(x)$)? Compared with additive perturbations, what are the theoretical or practical advantages of this approach, and were alternative strategies considered?

---

> ### Author Response · Authors · 2025-11-21
>
> Thank you for your thoughtful comments and time. We respond to specific comments below.
>
> - **Assumptions and proof of Theorem 2**
> We detail in the paper the assumptions of Theorem 2:
> $\hspace{1cm}$ $\rightarrow$ Threat model: adversary that has full knowledge of our algorithm, hyperparameters and is computationally unbound.
> $\hspace{1cm}$ $\rightarrow$ Theorem 2 is a simple union bound argument (Bonferroni correction) on top of Lemma 1 as specified in Appendix G.1
> $\hspace{1cm}$ $\rightarrow$ Lemma 1 assumes that the preference signals $r_j$ are independent and identically distributed as Bernoulli variable $\mathcal{B}(p)$, where in practice p = $\frac{1}{s}\sum_{j=1}^s r_j$.
> We would be happy to clarify any additional proof steps.
>
> - **Finite branching factor**
> The branching factor represents the number of possible comparison outcomes at each iteration, which is usually = 2 when one compares the current model to the best found so far. The Gaussian noise sampling is fixed under the given seed $\omega$ (the algorithm is deterministic given $\omega, D, a, b$), and a random algorithm is obtained through randomizing the seed $\omega$. We would gladly provide more details if this doesn't fully answer your question.
>
>
> - Thank you for pointing out the typo concerning $\pi_i k_i$, it is corrected in the updated version to $\prod_{i=1}^b k_i$ which represents the number of possible states after b iterations and $k_i$ the branching factor at iteration i.
>
>
> - We include all the libraries used and experimental details in the paper and Appendix H. We would be glad to clarify and include any missing details.
>
>
> - **Experimental scope**
> We agree with the reviewer that we do not evaluate our method on a wider range of tasks. This is primarily because the focus of our paper is theoretical. We conducted additional experiments on ARC Easy, ARC Challenge and Hellaswag. See discussion with reviewer *k5Ai* for a full exposition. We include these additional experiments in the revised version of the paper in Appendix I.
>
>
> - **Privacy claim**
> Please see discussion with reviewer *kVTM* for additional details.
> In the paper we distinguish between 2 cases:
> $\hspace{1cm}$ 1- When users provide their preference signal and it is not treated as private information, BBoxER requires only this signal and does not access the users’ input prompts or outputs. This design inherently ensures privacy: the prompts and outputs can be altered without affecting BBoxER’s results.
> $\hspace{1cm}$ 2-The general case when we need to compute the preference signal or if we consider that the user provided preference signal is a private information. In this scenario the provided bound results from Theorem 2 with $m=1$ providing $(\epsilon=0, \delta=3b/\sqrt{2\pi s})$, and we agree that it is insufficient to provide a meaningful bound from a DP perspective, which we state clearly in the updated version.

---

> > ### Author Response · Authors · 2025-11-21
> >
> > - **Robustness to extraction and MIA**
> > We provide theoretical guarantee and intuition of how the strong compression of BBoxER makes it unlikely for extraction attacks to work in section 4.3. Given that the entirety of the dataset is compressed to a comparison trace of few bits, the same trace could be obtained from various and different datasets making it very unlikely to identify the training dataset. Experiment 4 in the paper addresses the robustness against MIA loss based attacks such as LiRA [1], where we show that BBoxER is more robust compared to SFT or DP-AdamW.
> >
> >
> > - We thank the reviewer for pointing out that some of the references were duplicated. We have corrected this in the updated version.
> >
> >
> > - **Exponential form of parameter update**
> > Our theory omits the choice of the $modified$ function, or in other words covers all possible choices of such function. We chose the exponential multiplicative update purely through an experimental process, as we've noticed early on that additive $modified$ function $w = w + 0.01 \times \varepsilon$ performed less well in experiments.
> > The choice to optimize a fraction of parameters instead of the entire model, is first backed by the intuition that optimizing a large number of parameters would require more iterations and we want to stay in the low iteration regime that is covered by our theory. And secondly, updating a fraction of parameters allows for faster iterations from a practical viewpoint.
> >
> > **References**
> >
> > - [1] Carlini, N., Chien, S., Nasr, M., Song, S., Terzis, A. and Tramer, F., 2022, May. Membership inference attacks from first principles. In 2022 IEEE symposium on security and privacy (SP) (pp. 1897-1914). IEEE.

---

> > > ### Comment · Reviewer_PoZh · 2025-11-26
> > >
> > > Thanks to the authors' efforts, it addresses my major concerns. But the reviews significantly diverge, so I need to consult other reviewers' responses to the rebuttal.

---

### Official Review · Reviewer_k5Ai · 2025-10-31

**Soundness:** 2
**Presentation:** 2
**Contribution:** 2
**Rating:** 4
**Confidence:** 3

**Summary:**

This paper introduces BBoxER, a framework for LLM post-training using gradient-free Black-Box Optimization (BBO). The method avoids gradients by using comparison-based algorithms, which compress the dataset into a sequence of binary comparison results . The authors leverage this information bottleneck to derive theoretical guarantees for generalization, differential privacy, and robustness to poisoning/extraction. Empirically, they show BBoxER improves reasoning on Llama3.1-8B and resists membership inference attacks.

**Strengths:**

1. Linking BBO compression to generalization bounds that are independent of parameter count is a significant theoretical contribution.
2. Paper provides formal theorems for robustness against poisoning and extraction.
3. Paper shows detailed empirical results.

**Weaknesses:**

1. The method is not scalable. It only "works" by tuning a trivially small subset of parameters. This is a minimal tweak, not a scalable framework. Does the privacy stems from the fact that it barely changes the model?
2. The DP guarantee seems to be meaningless. The "general case" privacy guarantee is $(\epsilon=0, \delta \propto b/\sqrt{s})$. Using the paper's own numbers ($b=1200, s=7473$), $\delta \approx 13.9$. A $\delta > 1$ is a meaningless guarantee.
3. The paper claims to be "memory-efficient"  but ignores the astronomical computational cost. Each step of the BBO algorithm requires at least one full evaluation pass over the training data to get a single comparison bit. Could authors show the detail of running time and memory usage of this part?
4. Only evaluate on limited number of datasets.

**Questions:**

See weaknesses

---

> ### Author Response · Authors · 2025-11-21
>
> We thank the reviewer for their comments and time. Below are our point-wise responses to each of your concerns:
>
> - **Scalability of the approach**
> Regarding the scalability concern, our method doesn't operate on a fixed number of parameters, but on a small *fraction* of the parameters, which therefore scales with model size. Similar approaches in the literature like PEFT (LoRA [1] for example) or REFT [2] also update only a fraction of parameters during training. It is also important to state that we get a notable improvement on top of instruct models even under this small parameter change regime. Furthermore, our framework is flexible: we can use any $modified$ function including changing the entire model weights. Our theoretical bounds and results on generalization, privacy of user inputs/outputs and robustness to poisoning and extraction attacks, benefit from a small number of iterations $b$. And generally, having more parameters would require more iterations to optimize, in addition to longer sampling, state processing from the BBO algorithm and weights update at each iteration slowing the runtime of BBoxER from a practical viewpoint.
> $\hspace{1cm}$$\rightarrow$ Our theoretical results on generalization, privacy, robustness to poisoning and extraction attacks are agnostic to the choice of $modified$ function : the guarantees are true for all functions that update the model only from the noise sampled from the BBO algorithm, and therefore the theoretical guarantees don't depend on the number of updated parameters.
>
> - **DP guarantees**
> Please see discussion with reviewer *kVTM* for additional details.
> We agree with the reviewer that the general case doesn't provide a sufficient bound. Our goal was to simply provide a link and intuition on the impact of dataset size $s$ and number of iterations $b$. Our initial and main contribution for privacy is in the A/B testing case where the user preference signal isn't a private information as it is the natural scenario to use a comparison based learning algorithm. In that case, BBoxER doesn't need access to the users input prompts and outputs providing perfect privacy by design.
> Following your remarks, in the updated version, we removed the mention of differential privacy in the second, more general, case and instead described it solely in terms of bounded influence. Additionally, we included a discussion linking the inequality to differential privacy while clearly stating that it is insufficient.

---

> > ### Author Response · Authors · 2025-11-21
> >
> > - **Memory efficiency and computational cost**
> > The claim of memory efficiency is relative to gradient based learning where we don't need to store the gradients or large optimizer states. As training an LLM generally requires 16 times more memory than the memory required to load the model on GPU (in bfloat16) while it requires only twice as much to deploy it for inference, making it 8 times more memory efficient.
> > In the paper we include the runtime of each iteration as well as the GPU setup used for the experiments in the paper in Appendix H. Below we detail runtime comparison between BBoxER and SFT on an example case, as mentioned with reviewer *chir* :
> > For a fair comparison we fix a setup where we have 8 H100 GPUs, Llama-8B as our base model and GSM8K as our training dataset:
> > $\rightarrow$ **SFT** using FSDP (as one 80GB GPU isn't enough for finetuning an 8B model) and the hyperparameters from [4] : global batch size of 16 (2 on each GPU) for 5 epochs.
> > $\hspace{1cm}$ $\Rightarrow$ This gives an average run time of each epoch at: 04 min 20 sec
> > $\rightarrow$ **BBoxER** for 5 iterations using norm updates and DiagonalCMA (as in Experiment 3 of the paper). Each GPU contains a copy of the model instantiated with vLLM [3] with a maximum of 512 sampled tokens. Each GPU processes $7343/8\approx 918$ samples at each iteration.
> > $\hspace{1cm}$ $\Rightarrow$ This gives an average run time of each iteration at: 00 min 30 sec
> > Note that for BBoxER, we generate the chain of thought and the answer for evaluation, and the iteration time would be even shorter if we were optimizing logprobabilities (no token sampling) instead of accuracy.
> >
> >
> > - **Additional evaluations**
> > We agree with the reviewer that we do not evaluate our method on a wider range of tasks. This is primarily because the focus of our paper is theoretical. In addition, we chose to evaluate only on tasks whose performance is measured with non-differentiable metrics, which better fit black box optimization. It is also worth noting that BBoxER is inherently unsuitable for knowledge intensive tasks that require memorizing facts, due to its strong compression.
> > We conducted additional experiments using Llama-3.1-8B under the same setup as Experiment 3 in the paper. We trained on the ARC-Easy training split and evaluated on the ARC-Easy test split, and repeated the same procedure for ARC-Challenge and Hellaswag. We report the mean and standard deviation across five random seeds of the relative improvement, or difference of accuracy, compared to the base model.
> >
> >
> > |   BBoxER budget b  | ARC Easy        | ARC Challenge    | Hellaswag       |
> > |------|-----------------|------------------|-----------------|
> > |  150 | 0.01 $\pm$ 0.10 | -0.14 $\pm$ 0.12 | 0.16 $\pm$ 0.07 |
> > |  300 | 0.13 $\pm$ 0.06 | -0.21 $\pm$ 0.17 | 0.25 $\pm$ 0.11 |
> > |  600 | 0.18 $\pm$ 0.16 | -0.05 $\pm$ 0.25 | 0.47 $\pm$ 0.06 |
> > | 1200 | 0.85 $\pm$ 0.16 | 0.24 $\pm$ 0.21  | 0.89 $\pm$ 0.13 |
> >
> >
> > These results illustrate how BBoxER behaves on tasks beyond mathematical reasoning. We notice that on ARC Easy and Hellaswag we get an improvement, while it is not the case on ARC Challenge. This likely stems from the base model’s performance across tasks. On ARC Easy and Hellaswag, the base model scores about 80%, so “retrofitting” can provide meaningful improvement. In contrast, on ARC-Challenge the base is about 50% (random is at ~25%), and BBoxER is designed to provide additional improvement as a lightweight add-on, rather than heavy duty training: if the base is weak, perturbing parameters won’t meaningfully improve the model even if we notice a slight improvement on the training split.
> > We include this additional experiment in Appendix I.
> >
> > **References**
> >
> > - [1] Hu, E.J., Shen, Y., Wallis, P., Allen-Zhu, Z., Li, Y., Wang, S., Wang, L. and Chen, W., 2022. Lora: Low-rank adaptation of large language models. ICLR, 1(2), p.3.
> >
> > - [2] Wu, Z., Arora, A., Wang, Z., Geiger, A., Jurafsky, D., Manning, C.D. and Potts, C., 2024. Reft: Representation finetuning for language models. Advances in Neural Information Processing Systems, 37, pp.63908-63962.
> >
> > - [3] Kwon, W., Li, Z., Zhuang, S., Sheng, Y., Zheng, L., Yu, C.H., Gonzalez, J., Zhang, H. and Stoica, I., 2023, October. Efficient memory management for large language model serving with pagedattention. In Proceedings of the 29th symposium on operating systems principles (pp. 611-626).

---

### Official Review · Reviewer_kVTM · 2025-11-01

**Soundness:** 1
**Presentation:** 1
**Contribution:** 2
**Rating:** 0
**Confidence:** 4

**Summary:**

The paper analyzes privacy and generalization error for post-training via comparison-based black-box optimization under discrete signal. They prove that due to the limited number bits contributed by each data record, such algorithm enjoys non-vacuous  generalization error bound that scale linearly with dataset size, and as corollary establish provable robustness to poisoning and resistance to extraction attacks, which are further confirmed by experiments of evaluating the proposed algorithm via Membership Inference Attacks.

**Strengths:**

- Extensive experiments showing that the proposed algorithm resist Membership Inference Attacks, unlike fine-tuned counterparts at equal utility.
- The problem of information leakage at post-training phase is timely.

**Weaknesses:**

- Lack of clarity: the main algorithm 1 has many undefined functionalities such as 'modified', 'tell', 'ask', and contains many completely terms $I_i, i=1, b$ that are unused after creation. Given the current state, it is almost impossible to precisely interpret what the algorithm executes as well as the novelty (which the authors claim in the introduction).
- Differential privacy guarantee in Section 4.2 is vacuous, meaningful DP requires delta to be polynomially small compared to dataset size (Vadhan 2017), while the DP guarantee in Section 4.2 is inverse proportional to sqrt(dataset_size), not strong enough to provide meaningful privacy guarantee. There is also no formal proof for the DP bound in the paper.
- No discussions on existing related works that show information leakage during post-training, under SFT and RL. (Such as Hayes 2024, Barbero 2025)


References:
- Vadhan, S. (2017). The complexity of differential privacy. In Tutorials on the Foundations of Cryptography: Dedicated to Oded Goldreich (pp. 347-450). Cham: Springer International Publishing.
- Hayes, J., Shumailov, I., Porter, W. P., & Pappu, A. (2024). Measuring memorization in RLHF for code completion. In The Thirteenth International Conference on Learning Representations.
- Barbero, Federico, Xiangming Gu, Christopher A. Choquette-Choo, Chawin Sitawarin, Matthew Jagielski, Itay Yona, Petar Veličković, Ilia Shumailov, and Jamie Hayes. "Extracting alignment data in open models." arXiv preprint arXiv:2510.18554 (2025).

**Questions:**

See weakness.

---

> ### Author Response · Authors · 2025-11-21
>
> We thank the reviewer for their time, and for finding that our method addresses a timely problem for LLM post-training. We respond to specific weaknesses below, and please let us know if you have any additional questions.
>
> - **Algorithm 1 explanation**
> We understand that the notation of Algorithm 1 might be uncommon at first sight, but the choice of using .ask and .tell is on purpose as it is the exact formulation used in the code by the *NeverGrad* package. A line-by-line explanation of the algorithm is provided alongside it, which we make clearer in the updated version of the paper. We also provide example $modified$ functions in Appendix B.
> The notation $I_i$ simply refers to the internal state of the algorithm at iteration $i$. It is not used subsequently since most of the results rely on the bound on the number of possible internal states over all possible datasets in Equation (4).
>
> Below we provide a simple execution of the BBoxER as an example, using *OneFifth* (algorithm 2, (1+1)-ES in Appendix D.1 in the paper) as the BBO algorithm and *Full Layer Update* on the normalization layers as our choice of $modified$:
>
> >Input: seed $\omega$, dataset D, BBO algorithm a=*OneFifth*, budget b
> *step 1* : initialize the internal state by seeding the rng with $\omega$, and fixing the step_size $\sigma$
> *step 2* : sample a $z \sim \sigma\mathcal{N}(0,I)$ via a.ask()
> *step 3* : update model weights $m_i = modified(m_0, z)$, where every RMSNorm layer weights $w_i^{(j)}$ are updated with $w_i^{(j)}=w_0^{(j)} \times exp{(0.01 \times z_j)}$, and $z = (z_1,...,z_\\text{number RMSNorm layers})$
> *step 4* : $k_i = a.numCases() = 2$ (we only compare with the best so far, therefore $2$ possible choices)
> *step 5* : $choice_i = a.compare((m_{best}, m_i), D)$ this comparison is either done by majority vote in A/B testing or by comparing the performance of $m_{best}$ and $m_i$ over D (note that only the comparison result is needed and not the actual performance of each model)
> *step 6* : $a.tell(choice_i)$ and $I_i = (I_{i-1}, choice_i)$ update the internal state of algorithm a : if $m_i$ is better ($choice_i=1$) then $\\sigma = 2*\\sigma$, otherwise ($choice_i=0$) $\sigma = 2^{-\\frac{1}{4}}*\\sigma$
> $\\rightarrow$ *repeat steps 2 to 6 for b iterations*
>
>
> We included this example in the updated version of the paper in Appendix K to improve the clarity of algorithm 1.
>
>
> - **Differential privacy claim**
> Regarding the differential privacy claim, we agree that the result for the privacy of user preference signals, where $\delta \sim b/\sqrt s$ isn't enough for a meaningful differential privacy guarantee. Our main privacy result is presented in the A/B testing setting which is the natural use case for comparison based learning. We assume that user $j$ provides a preference signal $r_j$ that is not considered private, whereas their input prompts and outputs warrant protection. In that scenario we guarantee perfect privacy for prompts and outputs as BBoxER only relies on the preference signal to work, meaning that changing the prompts and outputs doesn't affect BBoxER output as long as the $r_j$ remains the same.
> Following your very relevant comments, we decided to remove the mention of differential privacy altogether in the second, more general, case and only keep a discussion linking the inequality to differential privacy and adding a note, as the reviewer stresses upon, that it is insufficient. Regarding the proof of the inequality itself, it is a simple corollary from the poisoning bound of Theorem 2, where the adversary can change at most 1 point (m = 1), this is detailed in Appendix F.1.
> On a side note that might be of interest, it is possible to render BBoxER differentially private in the general case by considering the noised average of user preferences (in case of binary signal) prior to computing $choice_i$ by adding a centered Gaussian noise at each iteration i where the variance of the noise can be worked backward from a target $(\epsilon,\delta)=(b*\epsilon_\text{single},b*\delta_\text{single})$ using the bound in [1] or the optimal calibration from [2], but this is beyond the scope of our paper.
> Indeed, we view our main contribution as introducing the comparison based BBO framework to LLM posttraining and providing non-vacuous generalization and robustness results, as well as showing that BBoxER indeed leads to learning and improvement.

---

> > ### Author Response · Authors · 2025-11-21
> >
> > - **Discussion on leakage for SFT and RL during post-training**
> > We thank the reviewer for the suggested references. We added a discussion of the material that is relevant to our work in the "background and related work" section in the updated version of the paper, but we would like to draw the reviewer’s attention to the fact that the reference (*Barbero, Federico, Xiangming Gu, Christopher A. Choquette-Choo, Chawin Sitawarin, Matthew Jagielski, Itay Yona, Petar Veličković, Ilia Shumailov, and Jamie Hayes. "Extracting alignment data in open models." arXiv preprint arXiv:2510.18554 (2025)*) first appeared on arXiv on 21 October 2025, which is significantly *after* our paper submission deadline.
> >
> >
> > We want to emphasize that our main contributions in the paper are not limited to the privacy guarantees, and that it is a subresult of a subsection in the paper. We highlight that our contributions extend well beyond this stability analysis, including: information theoretic bounds derived from the finite branching factor of comparison based BBO, robustness guarantees to poisoning and extraction, and empirical improvements on LLM reasoning.
> >
> >
> > We hope that these clarifications address your concerns, and we would of course be grateful if you felt they justified a reconsideration of your score.
> >
> > **References**
> >
> > - [1] Dwork, C., & Roth, A. (2014). The Algorithmic Foundations of Differential Privacy. Foundations and Trends® in Theoretical Computer Science.
> >
> > - [2] Balle, B. and Wang, Y.X., 2018, July. Improving the gaussian mechanism for differential privacy: Analytical calibration and optimal denoising. In International conference on machine learning (pp. 394-403). PMLR.

---

### Official Review · Reviewer_chir · 2025-11-01

**Soundness:** 3
**Presentation:** 4
**Contribution:** 2
**Rating:** 6
**Confidence:** 4

**Summary:**

This paper introduces BBoxER, a framework that uses comparison-based black-box optimization (BBO) for retrofitting models without requiring gradient access. BBoxER's core contribution is creating an information bottleneck by compressing the entire dataset into a small optimization trace of comparison outcomes. This compression mechanism allows the paper to provide strong theoretical guarantees for differential privacy, robustness to data poisoning. Experiments on LLMs demonstrate that BBoxER improves reasoning performance and empirically resists MIAs, proving it a practical method for achieving safe adaptation without sacrificing utility.

**Strengths:**

1. This work is the first to formally connect the finite "branching factor" of BBO algorithms to quantifiable safety guarantees (like generalization, privacy) for LLMs.
2. The paper is well-written and conceptually clear.

**Weaknesses:**

1. The paper emphasizes memory efficiency (no gradients) but lacks a critical discussion of computational cost. The BBO approach requires $b$ iterations (up to 1200 in experiments), with each comparison involving a full pass over the training dataset. This implies a potentially massive computational cost compared to standard SFT (2-5 epochs).
2. The MIA robustness claim relies on a proxy metric (NLL difference), which is not an appropriate evaluation metric for MIA. It is recommended to use standard MIA methods such as LiRa [1] or other methods designed for LLMs [2,3,4], and report common MIA metrics such as TPR at low FPR.
3. The paper appears to directly leverage an existing BBO algorithm for post-training on LLMs. This is not a novel framework, and the authors are encouraged to reframe their contribution accordingly.

**Questions:**

What is the true computational cost of BBoxER compared to standard SFT?

---

> ### Author Response · Authors · 2025-11-17
>
> We thank the reviewer for their comments and time. We are currently working to provide an answer to the mentionned weaknesses and questions.
>
> We could assume that the reference to the LiRA attack is the following (*Nicholas Carlini, Steve Chien, Milad Nasr, Shuang Song, Andreas Terzis, and Florian Tramer. Membership inference attacks from first principles. In 2022 IEEE symposium on security and privacy (SP), pp. 1897–1914. IEEE, 2022.*). However, we believe that the citations [2, 3, 4] mentioned in the review may have been omitted.
> Would you kindly include these citations so we can correctly address your concerns?

---

> ### Author Response · Authors · 2025-11-21
>
> We thank the reviewer for the positive feedback regarding the clarity and quality of our writing. In the following we address the mentioned weaknesses and questions:
>
> - **Computational cost of BBoxER compared to SFT**
> Comparing the computational cost of SFT and BBoxER is challenging because they operate under different settings. SFT evaluates the logprobability of the answer given the prompt and chain of thought, while in our experiments with BBoxER, we sample both the chain of thought and the answer, using accuracy as our primary metric to select $choice_i$. Although it is possible to use logprobabilities for model comparison in BBoxER, our focus is on optimizing accuracy for reasoning benchmarks.
> Our claim is that BBoxER uses far less memory (around 8 times less memory for inference with LLMs compared to training) as we don't need to store gradients and large optimizer states which allows for fitting large batches and processing them at once.
> For the sake of comparison we fix a setup where we have 8 H100 GPUs, Llama-8B as our base model and GSM8K as our training dataset:
> $\rightarrow$ **SFT** using FSDP (as one 80GB GPU is not enough for finetuning an 8B model) and the hyperparameters from [4] : global batch size of 16 (2 on each GPU) for 5 epochs.
> $\hspace{1cm}$ $\Rightarrow$ This gives an average run time of each epoch at: 04 min 20 sec
> $\rightarrow$ **BBoxER** for 5 iterations using norm updates and DiagonalCMA (as in Experiment 3 of the paper). Each GPU contains a copy of the model instantiated with vLLM [3] with a maximum of 512 sampled tokens. Each GPU processes $7343/8\approx 918$ samples at each iteration.
> $\hspace{1cm}$ $\Rightarrow$ This gives an average run time of each iteration at: 00 min 30 sec
>
> Note that for BBoxER, we generate the chain of thought and the answer for evaluation, and the iteration time would be even shorter if we were optimizing logprobabilities (no token sampling) instead of accuracy.
> We also provide the runtime of each iteration as well as the GPU setup used for the experiments in the paper in Appendix H.
>
> - **MIA robustness**
> Regarding the MIA robustness, we propose to look at the sum of **absolute** NLL differences as most MIA attacks (including LiRA [1]) perform tests that can be formulated as follows: $f (NLL\\{sample\\}) > threshold\\{sample\\}$
> where f denotes a transformation that is usually continuous. We include in Appendix H an exact derivation for LiRA under the formulation above.
> It is interesting to look at this metric for the following reasons:
> 1- If the base model was pre-trained on certain samples using a detectable method, BBoxER cannot make it robust against such attacks. Therefore, it is more meaningful to measure how much the loss for each sample changes compared to the base model. Accordingly, we look at the absolute difference between the loss of the updated model and the base model over each sample. We take the absolute value so that negative and positive differences don't compensate each other, although in practice the loss of the updated model will most likely be lower than the base model since it has been trained on those samples.
> 2- There are many MIA attacks and their effectiveness can vary significantly depending on the used parameters. For example, in LiRA, even changes of the hyperparameters of the optimizer during training of shadow models can affect detection results [1]. Instead of relying on metrics sensitive to these parameters, it is preferable to use an intrinsic metric like the absolute NLL difference.
>
> Our claim is that if this metric is lower for BBoxER compared to other approaches, then BBoxER is more difficult to attack and thus more robust. We provide further intuition and a theoretical guarantee in Section 4.3: due to the strong compression of BBoxER (the entire dataset is compressed to a trace of few bits), the same optimization trace (and therefore the same model) can be obtained from various and very different datasets making it extremely unlikely to determine the training data.
>
>
> - **Contribution**
> We agree with the reviewer that we don't propose any new black-box optimization algorithm and instead leverage the existing comparison based ones and put them to use for LLM post-training by updating model weights. To our knowledge we are the first to do so, opening an avenue that allows to transfer the decades-long research from the BBO community to LLM post-training in the way we describe. We made this distinction clearer in the updated version.

---

> > ### Author Response · Authors · 2025-11-21
> > **References**
> >
> > **References:**
> > - [1] Carlini, N., Chien, S., Nasr, M., Song, S., Terzis, A. and Tramer, F., 2022, May. Membership inference attacks from first principles. In 2022 IEEE symposium on security and privacy (SP) (pp. 1897-1914). IEEE.
> >
> > - [2] Muennighoff, N., Yang, Z., Shi, W., Li, X.L., Fei-Fei, L., Hajishirzi, H., Zettlemoyer, L., Liang, P., Candès, E. and Hashimoto, T.B., 2025, November. s1: Simple test-time scaling. In Proceedings of the 2025 Conference on Empirical Methods in Natural Language Processing (pp. 20286-20332).
> >
> > - [3] Kwon, W., Li, Z., Zhuang, S., Sheng, Y., Zheng, L., Yu, C.H., Gonzalez, J., Zhang, H. and Stoica, I., 2023, October. Efficient memory management for large language model serving with pagedattention. In Proceedings of the 29th symposium on operating systems principles (pp. 611-626).

---

### Author Response · Authors · 2025-11-21
**General Response to all ACs and Reviewers**

# General Response to all ACs and Reviewers

We thank the reviewers for their thoughtful feedback and valuable points and are pleased that they acknowledged the strengths of the paper, mainly:


- **From compression to generalization, independently from model parameters**: reviewers highlighted the theoretical novelty "This work is the first to formally connect the finite ‘branching factor’ of BBO algorithms to quantifiable safety guarantees" [chir], and the significance of our contribution "Linking BBO compression to generalization bounds that are independent of parameter count is a significant theoretical contribution" [k5Ai].


- **Formal theorem and empirical results for robustness against poisoning and extraction**: reviewers emphasized both formal "Paper provides formal theorems for robustness against poisoning and extraction" [k5Ai] and empirical evidence "Extensive experiments showing that the proposed algorithm resist Membership Inference Attacks, unlike fine-tuned counterparts at equal utility" [kVTM] of the robustness of our approach.


- **Experiments validate the effectiveness of BBoxER**: reviewers commented on the "detailed empirical results" [k5Ai], and noted "accuracy gains under a no-gradient setting, confirming the method’s effectiveness" [PoZh].

In response to reviewers feedback and suggestions, we made the following changes and clarifications:

1- **Algorithm 1 clarity and Computational cost.**
Clarified the memory claim alongside the memory efficiency in the paper thanks to reviewers *k5Ai* and *chir* suggestions.
Clarified Algorithm 1 in section 3.1 and added a step by step example in a new Appendix K.

2- **Privacy claim.**
Based on reviewer feedback (*kVTM, k5Ai, PoZh*), we reorganized Section 4.2 to clearly distinguish two settings:
- The A/B-testing scenario, where BBoxER relies only on aggregated preference bits. In this setting, prompts and outputs are never accessed, yielding a strong and exact (0,0)-DP guarantee, considering prompts and outputs are private. This is now explicitly identified as our main privacy result.
- The general setting, where the earlier differential privacy claim is now reframed as a stability bound, with the text clearly noting its insufficiency for meaningful DP.

3- **New experiments on different tasks.**
Based on reviewer (*k5Ai, PoZh*) comments, we performed additional experiments on different tasks than mathematical reasoning and included them in Appendix I.

4- **Related work and cleanup.**
Added related work on leakage in post-training, corrected typos and notation, cleaned up duplicated references thanks to reviewer *PoZh*, and adjusted our contribution claim as suggested by reviewer *chir*.

All revisions are highlighted in blue in the updated paper. We appreciate the reviewers feedback, which substantially improved clarity, theoretical and empirical context, and we believe the paper now more clearly positions BBoxER as a practical, and theoretically grounded framework for safe and robust comparison based post-training of LLMs.

---

### Meta-Review · Area_Chair_ih65 · 2026-01-07

**Summary:**

The paper proposes BBoxER, a framework using comparison-based black-box optimization for LLM post-training to create an information bottleneck, theoretically offering privacy and robustness guarantees. While the perspective of linking BBO branching factors to safety is novel, the paper suffers from critical theoretical limitations regarding privacy guarantees and practical scalability concerns.

**Reviewer Concerns:**

The authors clarified the algorithm's notation and provided runtime comparisons, addressing presentation issues. However, the critical concern regarding the "vacuous" differential privacy guarantee in the general case was conceded by the authors (who removed the claim), and concerns regarding the prohibitive computational cost of full-dataset evaluations for minimal parameter updates remain outstanding.

**Reviewer Scores:**

Reviewers kVTM and k5Ai would likely maintain their rejection stances given the confirmed lack of meaningful DP guarantees. Reviewer chir might lower their score slightly due to the clarified but still high computational overhead, while PoZh might lower their score upon seeing the theoretical gaps identified by others, converging on a borderline rejection.

Vacuous Differential Privacy Guarantees: Reviewers identified that the derived privacy bounds for the general case were meaningless (scaling inversely with dataset size), a flaw the authors conceded by removing the claim in the revision.

Prohibitive Computational Cost: The method requires a full pass over the training dataset for every single comparison iteration, making it computationally inefficient compared to standard SFT despite claims of memory efficiency.

Limited Scalability and Scope: The approach is demonstrated only by tuning a very small fraction of parameters or specific layers, raising doubts about its ability to scale to full-model adaptation or complex tasks beyond simple reasoning.

Incremental Novelty: The core contribution is largely the application of off-the-shelf comparison-based BBO algorithms to LLMs, and without the strong privacy proofs initially claimed, the theoretical novelty is significantly diminished.

---

### Decision · Program_Chairs · 2026-01-26

Reject